# BALLAST: Bayesian Active Learning with Look-ahead Amendment for Sea-drifter Trajectories under Spatio-Temporal Vector Fields

Rui-Yang Zhang [1]    Lachlan Astfalck [2 3]    Edward Cripps [2]    David Leslie [1]    Henry Moss [1]

## Abstract

We introduce a formal active learning methodology for guiding the placement of Lagrangian observers to infer time-dependent vector fields – a key task in oceanography, marine science, and ocean engineering – using a physics-informed spatio-temporal Gaussian process surrogate model. The majority of existing placement campaigns either follow standard 'space-filling' designs or relatively ad-hoc expert opinions. A key challenge to applying principled active learning in this setting is that Lagrangian observers are continuously advected through the vector field, so they make measurements at different locations and times. It is, therefore, important to consider the likely future trajectories of placed observers to account for the utility of candidate placement locations. To this end, we present BALLAST: Bayesian Active Learning with Look-ahead Amendment for Sea-drifter Trajectories. We observe noticeable benefits of BALLAST-aided sequential observer placement strategies on both synthetic and high-fidelity ocean current models. In addition, we developed a novel GP inference method – the Vanilla SPDE Exchange (VaSE) – to boost the GP posterior sampling efficiency, which is also of independent interest.

## 1. Introduction

Understanding and predicting ocean currents is of vital importance to mapping the flow of heat, nutrients, pollutants and sediments in the ocean (Ferrari & Wunsch, 2009; Keramea et al., 2021). Ocean currents are inferred from a plurality of measurement devices, such as fixed-location buoys, satellites and free-floating buoys, known as **drifters** (Lump-

kin et al., 2017). Free-floating drifters are being increasingly used due to their ability to sample both spatial and temporal flow properties and remain relatively affordable as compared to other measurement devices (Ponte et al., 2024). Once placed, drifters will be advected by the underlying (time-dependent) vector fields and take velocity measurements at different locations and times, thus they are **Lagrangian observers** since they represent the Lagrangian specification of flows (Griffa et al., 2007).

The majority of existing drifter placement campaigns either follow standard 'space-filling' designs (Tukan et al., 2024) or relatively ad-hoc expert opinions (Van Sebille et al., 2021; Poje et al., 2002). There also exists work such as Salman et al. (2008); Chen et al. (2024b); Bollt et al. (2024) that proposed hand-crafted criteria (e.g. travel distance and placement separation) for placement under the Lagrangian data assimilation inference framework (Apte et al., 2008) that appeal to information theory. However, a placement strategy explicitly using active learning, to the best of our knowledge, has not yet been presented in the literature.

Active learning (Settles, 2009) is a type of sequential experimental design (Gramacy, 2020) that iteratively selects the optimal observation point to maximise the total knowledge about the system of interest given existing data by optimising a utility function — often related to the information gain of the observation outcome (Rainforth et al., 2024). As the system of interest here is the evolving ocean currents, we consider the *spatio-temporal* active learning over a two-dimensional spatial region and a finite time horizon.

After realising the inadequacy of standard active learning methods for Lagrangian observers (see Section 3.1 for more details), we propose **BALLAST** — Bayesian Active Learning with Lookahead Amendment for Sea-drifter Trajectories. BALLAST accounts for the data structure of Lagrangian observers by simulating hypothetical trajectories using vector fields. Here, we used a spatio-temporal vector-output Gaussian process (GP) surrogate (see Figure 1 for an illustration) and proposed an information-theoretic utility for the active learning, and devised an original GP inference method, the **Vanilla SPDE Exchange** (VaSE), for efficient posterior sampling that is *thousands* of times faster than SPDE-GP and *billions* of times faster than standard GP for

[1]Lancaster University [2]University of Western Australia [3]University of New South Wales. Correspondence to: Rui-Yang Zhang <r.zhang26@lancaster.ac.uk>.

*Proceedings of the 43rd International Conference on Machine Learning*, Seoul, South Korea. PMLR 306, 2026. Copyright 2026 by the author(s).

the problems we consider here (see Section 4.1 for details).

Our contributions can be summarised as follows: (i) we introduce active learning concepts to the literature of Lagrangian observer placements, (ii) we propose BALLAST, a novel active learning amendment that accounts for Lagrangian observations using samples from surrogates, and (iii) we develop the vanilla-SPDE exchange, a new GP inference method combining standard GP regression and the SPDE approach (Sarkka et al., 2013), for efficient BALLAST utility computation, which may be of independent interest especially for spatio-temporal GPs with non-gridded observations. Our numerical results suggest noticeable benefits of BALLAST-aided active learning for sequential observer deployment on both synthetic and high-fidelity ocean current models.

### 1.1. Notation

In the rest of the paper, we use $f$ to denote the object of interest with distribution $p(f)$. The object $f$ is usually modelled using a zero-mean and kernel $k$ GP, denoted by $f \sim \mathcal{GP}(0, k)$. The Gram matrix constructed with kernel $k$ is denoted by $K$. When the Gram matrix is computed between two identical test points $X$, i.e. $K(X, X)$, we will simplify the notation by $K(X) := K(X, X)$. We also use $p(f|\mathcal{D}_n)$ to denote the posterior distribution and $p(y|\mathcal{D}_n, x)$ to denote the posterior predictive distribution at $x$ after observing data $\mathcal{D}_n$. Samples from $p(f|\mathcal{D}_n)$ are denoted by $F$ in general and $F^{(j)}$ for the $j$-th sample.

## 2. Background

### 2.1. Gaussian Process for Time-Dependent Vector Fields

A surrogate model is employed in active learning to model the unknown object of interest $f$ while providing uncertainty quantification, commonly chosen to be a GP (Williams & Rasmussen, 2006; Gramacy, 2020). In this study, the system of interest is a time-dependent vector field, so we consider a spatio-temporal, vector-output GP as the surrogate used for active learning.

Following Ponte et al. (2024), we consider an extension to the Helmholtz kernel $k_{\text{Helm}}$ of Berlinghieri et al. (2023) – a vector-output kernel (Alvarez et al., 2012) using the Helmholtz decomposition (Bhatia et al., 2012) to more realistically portray the vector field structure – by including a separable temporal kernel $k_{\text{time}}$, which results in the temporal Helmholtz kernel

$$k_{\text{tHelm}}\left((\boldsymbol{s}, t), (\boldsymbol{s}', t')\right) = k_{\text{Helm}}(\boldsymbol{s}, \boldsymbol{s}')k_{\text{time}}(t, t')$$

with $\boldsymbol{s}, \boldsymbol{s}' \in \mathbb{R}^2, t, t' \in \mathbb{R}$. The spatial Helmholtz kernel $k_{\text{Helm}}$ is constructed from linear differential operators applied to the potential and stream function kernels, which are two two-dimensional input, scalar output kernels such as

the SE kernel (see Section B.2 for details). The temporal kernel $k_{\text{time}}$ is set to be a Matérn 3/2 kernel: evidence in oceanographic research suggests that the smoothness $\nu$ is around 2; as this leads to a non-analytic representation of the Bessel function, often $\nu = 3/2$ is taken as a sufficient approximation (Lilly et al., 2017; Ponte et al., 2024). We will also use $\boldsymbol{x} = (\boldsymbol{s}, t)$ to denote the input of the GP. An illustration of the spatio-temporal GP regression of Lagrangian trajectories is displayed as Figure 1.

### 2.2. Sequential Experimental Design

Sequential experimental design, with active learning as a special case, selects an optimal measurement point $x_n^*$ from measurement set $X$ at each time $t_n$ using existing data $\mathcal{D}_n$ by optimising the expectation of utility function $U$ over the posterior predictive distribution $p(y|\mathcal{D}_n, x)$ at $x \in X$, mathematically formulated as

$$x_n^* = \operatorname{argmax}_{x \in X} \mathbb{E}_{y \sim p(y|\mathcal{D}_n, x)}[U(y)]. \qquad (1)$$

A common utility choice for (Bayesian) active learning is the negative entropy (see Section B.3 for definitions) (Lindley, 1956; Ryan et al., 2016). For a probabilistic object of interest $f$, the **information gain** of an additional observation $y$ given existing data $\mathcal{D}_n$ is the reduction in entropy $H(\cdot)$ between the prior $p(f|\mathcal{D}_n)$ and posterior $p(f|\mathcal{D}_n, y) \propto p(f|\mathcal{D}_n)p(y|f)$, given by

$$IG(y) := H(p(f|\mathcal{D}_n)) - H(p(f|\mathcal{D}_n, y)).$$

Therefore, the **expected information gain** (EIG) policy selects the next measurement point $x_n^*$ using the posterior predictive $p(y|\mathcal{D}_n, x)$ at $x$

$$\begin{aligned} x_n^* &= \operatorname{argmax}_{x \in X} \mathbb{E}_{y \sim p(y|\mathcal{D}_n, x)}[IG(y)] \\ &= \operatorname{argmax}_{x \in X} \mathbb{E}_{y \sim p(y|\mathcal{D}_n, x)}[-H(p(f|y, \mathcal{D}_n))]. \end{aligned}$$

### 2.3. Problem Setup

Here, our active learning object of interest $f$ depends on both space and time. We assume pre-determined decision times, so we only select the placement location of the Lagrangian observers at each decision time. This simplifying assumption is commonly imposed in the literature to reduce the sequential design's search space (Huang et al., 2022; Fanshawe & Diggle, 2012; Wikle & Royle, 1999), and is an accurate representation of many real-world drifter deployment practices (Lilly & Pérez-Brunius, 2021). Additionally, we assume the Lagrangian observers produce data with negligible delay – a realistic assumption as modern surface drifters broadcast GPS positions in real time at regular intervals with high accuracy (Novelli et al., 2017).

In particular, we will focus on two-dimensional, time-dependent vector fields that are temporally defined on the

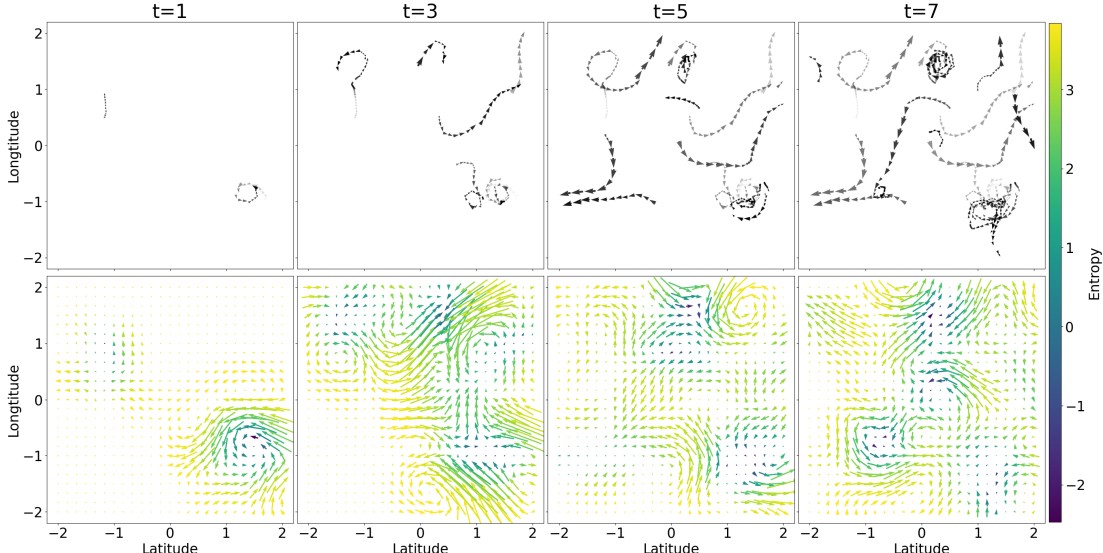

*Figure 1.* Illustration of spatio-temporal GP regression of Lagrangian trajectories. The top row shows the aggregated observations at different times. The bottom row shows the regressed GP marginals at corresponding times, where the posterior mean is plotted with colours following entropies of the respective random vectors.

finite time interval $[0, T]$ with terminal time $T \in \mathbb{R}$ and are spatially supported on a closed rectangle $[a_1, b_1] \times [a_2, b_2] \subset \mathbb{R}^2$. For some calculations, we will discretise the time interval into $N_{\text{time}}$ even segments with time steps $\mathcal{T} := \{0, \delta_t, \ldots, (N_{\text{time}}-1)\delta_t\}$, while the spatial domain is discretised into a regular grid with cells centred at $R = \{s_i\}_{i=1}^{N_{\text{space}}}$.

## 3. Active Learning of Time-Dependent Vector Fields

We are ready to present our active learning loop for identifying a time-dependent vector field under the setup outlined in Section 2 using the GP surrogate defined in Section 2.1. We model the target vector field using the temporal Helmholtz surrogate $f \sim \mathcal{GP}(0, k_{\text{tHelm}})$, and assume we make noisy velocity observations $y$ at observation location $x = (s, t)$ where

$$y = f(x) + \varepsilon = f(s, t) + \varepsilon, \qquad \varepsilon \sim N(0, \sigma_{\text{obs}}^2 I_2)$$

and $\sigma_{\text{obs}}$ is the standard deviation of the observation noise. The active learning selects the placement location of Lagrangian observers at each placement time in order to maximise the information gain of the surrogate $f$ over the full spatio-temporal grid.

Let $t_n$ be the time when we are deciding the placement location of the $n$-th observer. The observations so far are denoted by $\mathcal{D}_n = \{X_n, y_n\}$ with $X_n, y_n$ being the full observation locations and values. Given these observations,

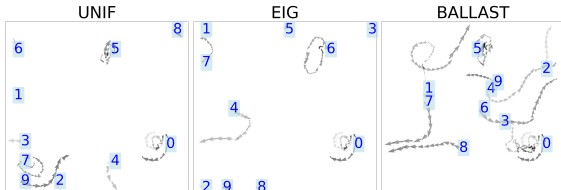

*Figure 2.* Deployment comparison under the uniform policy (left), EIG (middle), and our proposed BALLAST (right). Ten Lagrangian observers are placed sequentially, with their placement locations in blue. The observations are plotted with varying brightness according to the observation time (later is brighter).

the next placement location $s_n^*$ at placement time $t_n$ is

$$s_n^* = \operatorname{argmax}_{s \in R} \\ \mathbb{E}_{y \sim p(y|\mathcal{D}_n, s, t_n)}[-H\left(p(f|\mathcal{D}_n \cup \{(s, t_n, y)\})\right)] \tag{2}$$

where $p(y|\mathcal{D}_n, s, t_n)$ is the posterior predictive distribution at $(s, t_n)$.

Although standard computation of expected information gain like (2) involves a computationally prohibitive cost (Ryan et al., 2016), we can compute our expected utility efficiently here using the properties of Gaussian distributions' entropies as well as the symmetry of mutual information.

In particular, the posterior predictive distribution of a GP over a spatio-temporal grid, i.e. $p(f(R \times \mathcal{T})|\mathcal{D})$, is a multivariate Gaussian and its entropy is directly computable analytically from the distribution's covariance matrix. Furthermore, this entropy has no dependency on the observation value, thus the expected entropy is identical to the entropy itself. Finally, using the symmetry of mutual information,

one can obtain an equivalent formulation of the information gain in (2) that is computationally cheaper than its original form. We remark that such a reformulation using mutual information symmetry has been applied to entropy search (Hennig & Schuler, 2012) to yield the more cost-efficient predictive entropy search (Hernández-Lobato et al., 2014).

Overall, we have

$$
\begin{aligned}
\boldsymbol{s}_n^* &= \operatorname{argmax}_{\boldsymbol{s} \in R} \mathbb{E}_{y \sim p(y|\mathcal{D}_n, \boldsymbol{s}, t_n)}[IG(y)] \\
&= \operatorname{argmax}_{\boldsymbol{s} \in R} \log \det \left( I + \sigma_{\mathrm{obs}}^2 K(X_n^{+\boldsymbol{s}}) \right)
\end{aligned} \tag{3}
$$

where $X_n^{+\boldsymbol{s}} = X_n \cup (\boldsymbol{s}, t_n)$ is the full observation points after the hypothetical evaluation at $(\boldsymbol{s}, t_n)$ and $K(X_n^{+\boldsymbol{s}})$ is the Gram matrix of kernel $k_{\mathrm{tHelm}}$ between the full observation points. The details of the above reformulation can be found in Section C.1.

### 3.1. The Pitfall of Standard Active Learning for Lagrangian Observers

Standard active learning, as formulated in (1), is inadequate for Lagrangian observers. Recall from Section 1 that our placed observers will continuously measure at different locations and times while being advected by the underlying vector field. This property, however, is ignored in the utility computation (e.g. (2) and (3), where only the initial observation location is considered). Thus, standard active learning is suboptimal, as stated in Proposition 1, which we formalise and prove in Section D.

**Proposition 1** (Informal). *For sequential experimental design where observers are Lagrangian, standard utility construction yields suboptimal decisions.*

As shown in Figure 2, the EIG policy places observers near the border, which would often leave the considered region quickly and yield few observations. Numerical experiments in Section 5 even suggest EIG is consistently worse than the uniform policy.

## 4. BALLAST

To faithfully measure the utility of a placed drifter, we propose **BALLAST**: Bayesian Active Learning with Lookahead Amendment for Sea-drifter Trajectories, a novel algorithm that adjusts the utility computation via look-aheads using vector fields sampled from posteriors.

Intuitively, when estimating the utility of a placement, we aim to capture the subsequent observations made by the observer. To do so, we simulate the future trajectory of a placed observer until the terminal time using vector fields sampled from the posterior as proxies for the ground truth.

For any utility function $U$, the BALLAST-aided acquisition function is provided by

$$
\boldsymbol{s}_n^* = \operatorname{argmax}_{\boldsymbol{s} \in R} \mathbb{E}_{F \sim p(f|\mathcal{D}_n)} \left[ \mathbb{E}\left[ U(P_F^T(\boldsymbol{s}, t_n)) \right] \right] \tag{4}
$$

where $P_F^T(\boldsymbol{s}, t_n)$ denote the projected trajectory until terminal time $T$ of an object in vector field $F$ initialised at location $\boldsymbol{s}$ and time $t_n$, and $F$ is a sampled (time-varying) vector field from the posterior distribution $f|\mathcal{D}_n$. Note that we will only use observations within the considered spatial region, and terminate the observers when they leave it.

To compute the acquisition in (4), we approximate the outer expectation over $F \sim p(f|\mathcal{D}_n)$ using the Monte Carlo method (Robert & Casella, 1999) by taking $J$ draws $F^{(1)}, F^{(2)}, \ldots, F^{(J)}$ from the posterior $p(f|\mathcal{D}_n)$ and computing the integrand individually. The choice of $J$ is a key tuning parameter, which we pick $J = 20$ as the default following our ablation study's result in Sections 5.1 and H.

Each projected trajectory $P_{F^{(j)}}^T(\boldsymbol{s}, t_n)$ with candidate placement location $\boldsymbol{s} \in R$ and sample field $F^{(j)}$ is a collection of observation locations and times that can be obtained using a numerical ODE solver (e.g. Euler's method, Süli & Mayers (2003)) by iteratively updating the locations with a stepsize $\delta_t$ using velocities of the vector field $F$. Specifically, the velocity at a spatial location will be that of the grid cell containing the location. We would also project existing observers at locations $\boldsymbol{s}_{\mathrm{existing}}$ to obtain trajectories $P_{F^{(j)}}^T(\boldsymbol{s}_{\mathrm{existing}}, t_n)$. We denote $P_{F^{(j)}}^T(\boldsymbol{s}_{\mathrm{ag}})$ to be the aggregated additional observations after placing an observer at $\boldsymbol{s}$ under the sample field $F^{(j)}$.

BALLAST amendment is compatible with any utility function. Here, we will consider the special case of the information gain utility function, which yields the following BALLAST-aided acquisition function

$$
\begin{aligned}
\boldsymbol{s}_n^* &= \operatorname{argmax}_{\boldsymbol{s} \in R} \mathbb{E}_{F \sim p(f|\mathcal{D}_n)} \\
&\quad \left[ \log \det \left( I + \sigma_{\mathrm{obs}}^2 K\left( X_n^{+P_F^T(\boldsymbol{s}_{\mathrm{ag}})} \right) \right) \right] \\
&\approx \operatorname{argmax}_{\boldsymbol{s} \in R} \frac{1}{J} \sum_{j=1}^{J} \\
&\quad \left[ \log \det \left( I + \sigma_{\mathrm{obs}}^2 K\left( X_n^{+P_{F^{(j)}}^T(\boldsymbol{s}_{\mathrm{ag}})} \right) \right) \right]
\end{aligned} \tag{5}
$$

with $F^{(1)}, \ldots, F^{(J)} \sim f|\mathcal{D}_n$ being the sampled posterior vector fields, $K(\cdot, \cdot)$ denoting the Gram matrix with kernel $k_{\mathrm{tHelm}}$ and $X_n^{+P_{F^{(j)}}^T(\boldsymbol{s})} := X_n \cup P_{F^{(j)}}^T(\boldsymbol{s})$ be the full observation points under sample field $F^{(j)}$.

Close inspections of (5) indicate the computational bottleneck of the optimisation is the posterior sampling of vector fields $F^{(1)}, \ldots, F^{(J)}$ over the full spatio-temporal grid. Standard GP posterior sampling scales cubically in test

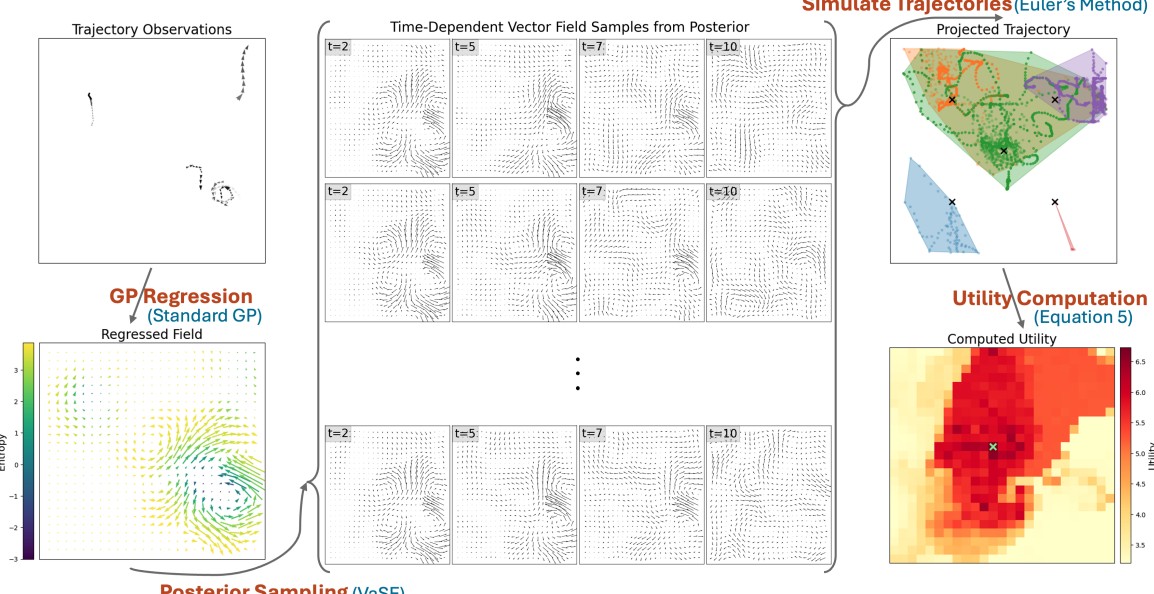

*Figure 3.* The schematic diagram illustrating the BALLAST algorithm for active learning. Given existing observations (top left), we first regress them using a GP (bottom left) and draw multiple samples from the posterior GP (middle). Hypothesised observation trajectories from candidate placements are simulated using sampled fields (top right), which are aggregated for utility computation (bottom right) to select the optimal deployment location (green cross in the bottom right plot).

points as it involves computing the matrix square root of a Gram matrix. Here, our test points consist of the full spatio-temporal grid of size $N_{samp} = N_{space}N_{sampT}$ – with $N_{space}$ spatial points and $N_{sampT}$ temporal points – commonly of the magnitude $10^5$ or higher, making standard sampling computationally impossible. In Section 4.1 below, we propose a novel GP inference method to overcome this bottleneck.

### 4.1. Vanilla-SPDE Exchange

To solve the challenge of posterior sampling for BALLAST, we propose the **Vanilla SPDE Exchange** (VaSE), a novel GP inference method that synergises the computational benefits of the standard GP and SPDE (Solin, 2016) frameworks, which is also of independent interest.

Under the SPDE-GP re-formulation, a 1D Matérn GP can be cast as the stationary solution of a linear SDE, which allows inference using the Kalman filter and Rauch-Tung-Striebel (RTS) smoother that costs linearly in time (Hartikainen & Särkkä, 2010). When the GP has a separable spatial kernel in addition to the Matérn temporal kernel, the same tools can be applied (Sarkka et al., 2013) and cost linearly in time and cubically in space. See Section F for more details.

While the SPDE approach of Sarkka et al. (2013) offers a computationally efficient re-formulation for spatio-temporal GPs, the method is not suitable for situations where the observation locations and test locations are minimally overlapping. Since the spatial component of the GP is included in

the SPDE using the Gram matrix of all considered locations, near-distinct observation and test locations yield a huge Gram matrix and cost cubically in size. As our Lagrangian observations are non-gridded, predicting on gridded test locations would thus yield prohibitive extra computational cost, making any potential speed-up of SPDE-GP futile. Details of this cost analysis can be found in F.4.

VaSE, on the other hand, uses standard GP regression to bypass the need to consider observation locations for SPDE-GP, so as to achieve efficient posterior sampling. We consider the extended GP $\boldsymbol{f} = [f, \partial_t f]^T$ with $f \sim \mathcal{GP}(0, k_{tHelm})$ and regress the observations with it. In particular, using the properties of kernels under linear operators (Agrell, 2019), the extended GP has kernel

$$\mathrm{Cov}((\boldsymbol{s}, t), (\boldsymbol{s}', t')) =$$
$$\begin{bmatrix} k_{tHelm}((\boldsymbol{s}, t), (\boldsymbol{s}', t')) & \partial_{t'} k_{tHelm}((\boldsymbol{s}, t), (\boldsymbol{s}', t')) \\ \partial_t k_{tHelm}((\boldsymbol{s}, t), (\boldsymbol{s}', t')) & \partial^2_{tt'} k_{tHelm}((\boldsymbol{s}, t), (\boldsymbol{s}', t')) \end{bmatrix}.$$

Since $k_{tHelm}$ is separable, the partial derivatives are only w.r.t. the Matérn temporal kernels.

To sample vector fields for BALLAST at time $t_n$ with observations $\mathcal{D}_n$ using VaSE, we first use the extended posterior distribution $\boldsymbol{f}|\mathcal{D}_n$ to generate the SPDE initial condition by drawing from $\boldsymbol{f}(R, t_n)|\mathcal{D}_n$, then propagate the initial condition until terminal time $T$ using the state space model. Full details of VaSE can be found in Section G.

The computational costs of drawing one posterior sample

| Method | Cost |
|---|---|
| Standard | $O(N_{obs}^3 + N_{pred,s}^3 N_{pred,t}^3 + N_{obs}^2 N_{pred,s} N_{pred,t} + N_{obs} N_{pred,s}^2 N_{pred,t}^2))$ |
| SPDE | $O((N_{obs} + N_{pred,s})^3 N_{obs,t} + N_{pred,s}^2 N_{pred,t})$ |
| VaSE | $O(N_{obs}^3 + N_{pred,s}^2 N_{obs} + N_{pred,s} N_{obs}^2 + N_{pred,s}^2 N_{pred,t})$ |

*Table 1.* Computational cost comparison of standard, SPDE, VaSE for drawing one posterior sample. We highlight the dominating terms $N_{pred,s}^3 N_{pred,t}^3$ for standard and $(N_{obs} + N_{pred,s})^3 N_{obs,t}$ for SPDE that make the two traditional methods drastically more expensive than VaSE. Full details in Section G.1.

using the three approaches (standard GP, SPDE-GP, vanilla-SPDE exchange) are compared in Table 1, where $N_{obs}$ ($\approx 200$) denotes the observation number, $N_{obs,t}$ ($\approx 200$) denotes the number of distinct observation times, $N_{pred,s}$ ($\approx 500$) and $N_{pred,t}$ ($\approx 1000$) denotes the number of prediction spatial and temporal points. Using the approximate values, we can notice that VaSE is of the cost order $10^8$, significantly cheaper than the $10^{11}$ cost of SPDE and $10^{17}$ cost of standard GP.

We also measured the wall run times of VaSE on a consumer laptop equipped with an Apple Silicon CPU and 24 GB RAM. In the same setup as that of Section 5.2, drawing one posterior sample with VaSE takes under 4 seconds (3.77s at decision time t = 2.5, 3.89s at t = 5.0, 3.64s at t = 7.5), whereas SPDE takes around 4.5 minutes (245.77s at t = 2.5, 259.83s at t = 5.0, 281.75s at t = 7.5). This additional comparison provides further evidence that VaSE is noticeably more efficient than the state-of-the-art SPDE approach for the kinds of sampling tasks considered in the paper.

### 4.2. BALLAST Algorithm

The BALLAST-aided active learning of time-dependent vector fields with Lagrangian observers using the expected information gain utility and a temporal Helmholtz GP surrogate is presented informally as Algorithm 1, presented formally in Section A, and visually represented as Figure 3.

The computational cost of one iteration of the BALLAST-EIG at time $t_n$ given $N_{obs}$ observations, $N_{space}$ spatial grid locations, and $N_{sampT}$ sampled time slices, is

$$O\left( \underbrace{J}_{\#\text{Samples}} \left( \underbrace{N_{obs}^3 + N_{obs} N_{space}^2}_{\text{Sample Field at } t_n} + \underbrace{N_{sampT} N_{space}^2}_{\text{Propagate Field}} \right.\right.$$
$$\left.\left. + N_{space} \left[ \underbrace{N_{sampT}}_{\text{Simulate Traj.}} + \underbrace{(N_{sampT} + N_{obs})^3}_{\text{Compute Utility}} \right] \right) \right)$$

where the blue indicates the complexity that can be reduced using parallelization. We also remark that under the setup considered in Section 5.2, it consistently takes under 3 min-

**Algorithm 1** BALLAST-EIG Active Learning of Lagrangian Observers (Informal)

1: **Input:** Deployment number $M$. BALLAST sample number $J$. Temporal Helmholtz GP $f$. Spatial grid $R$. Terminal time $T$.
2: Initialise an observer randomly at time $t_0 = 0$.
3: **for** $m = 1, 2, \ldots, M$ **do**
4:     Optimise GP hyperparameters using observations.
5:     **for** $j = 1, 2, \ldots, J$ *(parallelizable)* **do**
6:         Sample posterior field at deployment time $t_m$ using standard GP regression.
7:         Propagate sampled field until terminal time $T$ using the SPDE approach.
8:         Simulate trajectories of existing observers.
9:         **for** $s \in R$ *(parallelizable)* **do**
10:           Simulate the trajectory starting at $s$.
11:         **end for**
12:     **end for**
13:     Aggregate the utility contributions from $J$ samples to obtain the next placement $s_m^*$ via (5).
14:     Initialise an observer at $s_m^*$.
15: **end for**

utes (wall time, measured on a consumer laptop equipped with an Apple Silicon CPU and 24 GB RAM) to make a deployment decision with our recommended posterior sample number $J = 20$. In particular, at decision time $t = 3.0$, it took 2 min 45 s; at decision time $t = 5.0$, it took 1 min 55 s; at decision time $t = 7.0$, it took 2 min 13 s.

Our proposed Algorithm 1 builds on a separable, spatio-temporal GP surrogate with the temporal kernel being Matérn for the execution of the SPDE sampling procedure described in Section 4.1. This could be weakened following the work of Solin (2016) on constructing SPDE formulations for a broader range of kernels. However, the core BALLAST mechanism of trajectory projection is compatible with any utility choice and active learning surrogate models. We also remark that we initialise the algorithm with a uniformly drawn location for generality and simplicity.

# 5. Experiments

After an ablation study empirically analysing the tuning parameter choice of the BALLAST policy, we investigate the effectiveness of BALLAST for Lagrangian observer placement under ground truth generated by the temporal Helmholtz GP surrogate and under the high-fidelity Stanford Unstructured Nonhydrostatic Terrain-following Adaptive Navier–Stokes Simulator (SUNTANS) Fringer et al. (2006).

Six active learning policies are compared: uniform (**UNIF**), Sobol (**SOBOL**), distance-separation (**DIST-SEP**) heuristic inspired from Chen et al. (2024b), **EIG** of (3), and BALLAST of (5) with optimised and true hyperparameters (denoted **BALLAST-opt** and **BALLAST-true**). The Sobol sequence is chosen to represent space-filling-inspired policies such as Tukan et al. (2024).

The deployment policy proposed by Chen et al. (2024b) works under the Lagrangian data assimilation inference framework, and considers two criteria: (1) "the drifters are deployed at locations where they can travel long distances", and (2) "place the drifters at locations that are separate from each other". As we are working under a different inference framework, we adapt their policy and compute the criteria using GP posteriors and BALLAST samples, which gives us the DIST-SEP policy. Implementation details of all considered policies can be found in Section I.3, and the Github repo for the codes used in the experiments can be found at `https://github.com/ShuSheng3927/BALLAST`.

In general, we have observed consistently superior performance of BALLAST policies against other considered policies, with the Sobol policy being comparable to BALLAST-opt (but worse than BALLAST-true) in one setting. Additionally, the advantage of BALLAST over other policies increases as more observers are deployed.

## 5.1. Ablation Study

To determine a suitable choice of BALLAST sample number $J$ of Algorithm 1, we conduct an ablation study investigating the change in utility gap of the placement decision as $J$ increases. While a large number of samples is usually used to accurately approximate integrals for standard Monte Carlo methods, since our goal is to find the maximising location $s \in R$ of the expected utility, a small number of $J$ is often sufficient, as we will see below.

We consider a synthetic ground truth vector field generated by a temporal Helmholtz model (same as Section 5.2), and the full deployment duration is $[0, 10]$. Three different decision times $t = 3, 5, 7$ are considered, where uniformly placed drifters are initially placed every $0.5$ time prior to the decision. At each decision time, the true expected utility is approximated using $J = 200$, and the percentage utility

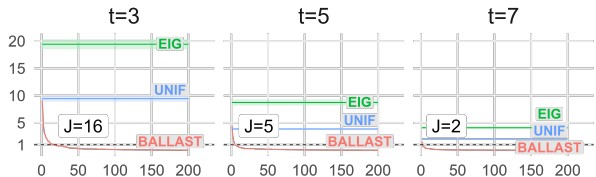

*Figure 4.* Percentage utility gap with 2 standard error bounds of Uniform, EIG, and BALLAST over posterior sample number $J$ at decision times $t = 3, 5, 7$. A percentage utility gap cut-off at $1\%$ is selected with corresponding $J$ values in text.

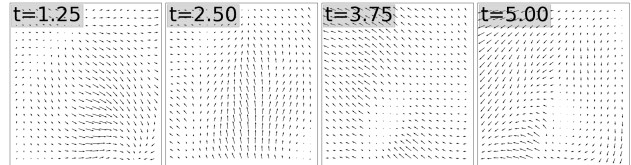

*Figure 5.* Vector fields at selected time slices of the SUNTANS dataset of Rayson et al. (2021).

gap between the optimal decision under $J = 1, 2, \ldots, 200$ against the optimal decision under $J = 200$ is calculated.

In Figure 4, BALLAST reached the $1\%$ utility gap before $J = 20$ consistently. Also, BALLAST decisions are consistently better than the uniform and EIG decisions for almost all choices of $J$, with EIG worse than Uniform – aligning with our observation in Figure 2. Full details of this ablation study and additional ablations can be found in Section H.

## 5.2. Temporal Helmholtz Ground Truth

Here, the ground truth vector field is drawn from a temporal Helmholtz GP described in Section 2.1 where the temporal kernel is a Matérn $3/2$ with lengthscale $2.5$ and variance $1.0$, and the Helmholtz kernel with two RBF kernels of variance $0.5$ and lengthscales $0.8$ and $0.5$ for potential and stream kernel respectively. The ground truth field is considered on a time grid $[0, 10]$ with time step $0.01$ and a spatial grid of size $25 \times 25$ evenly-spread on $[-2, 2] \times [-2, 2]$.

All policies (except SOBOL) are initialised uniformly at time zero, and 19 further observers are deployed every $0.5$ unit of time afterwards. The performance is measured by the average L2 error of the vectors of the posterior predictive mean field over the spatial grid and the full set of deployment times. For the experiment, 10 different sampled ground truth vector fields with 10 independent runs each are conducted. We compare the policies using the average policy rank and the iso-performance with the uniform policy as the benchmark. The policy rank considers the ranks of policies (one being the best) for each iteration, and the iso-performance considers the additional (positive or negative) number of observers needed to reach the same level of performance, averaged over each iteration's results.

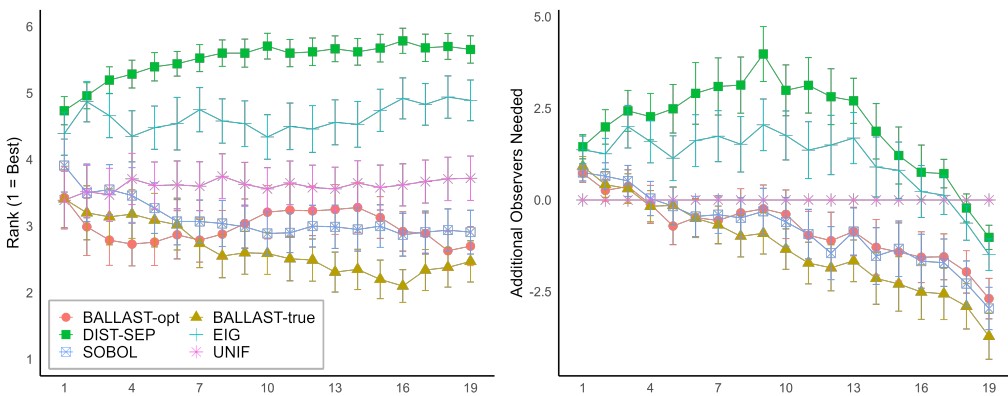

*Figure 6.* Policy comparison with temporal Helmholtz ground truth. Left is the average policy rank over iterations at each deployment time, with 2 standard errors. Right is the iso-performance over iterations with 2 standard errors.

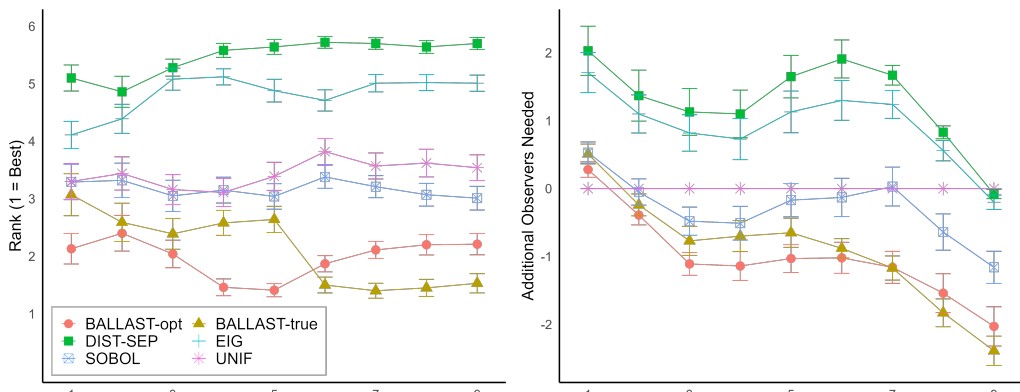

*Figure 7.* Policy comparison with SUNTANS ground truth. Left is the average policy rank over iterations at each deployment time, with 2 standard errors. Right is the iso-performance over iterations with 2 standard errors.

The result in Figure 6 indicates that BALLAST with true or optimised hyperparameters consistently outperforms all other policies, except for the Sobol sequence, which is worse than BALLAST-true and comparable with BALLAST-opt. At the end of the deployment, the two BALLAST policies save about 3 drifters against the uniform benchmark, which yields around $16\%$ deployment cost saving.

## 5.3. SUNTANS Ground Truth

The SUNTANS model is a high-fidelity numerical fluid mechanics model for non-hydrostatic flows (Fringer et al., 2006) and internal waves (Walter et al., 2012). Here, we use a (spatial and temporal) portion of the simulated, open-sourced vector fields from Rayson et al. (2021) as the ground truth vector field – see Figure 5 for an illustration. A temporal Helmholtz GP continues to be the surrogate model of choice here for the active learning policies.

For EIG and BALLAST-true policies in particular, as they require pre-specified hyperparameter values for the GP, we set the temporal kernel to be of lengthscale 1 and variance

15, the potential component of the Helmholtz kernel to be an RBF with lengthscale 5 and variance 20, and the stream component to be an RBF with lengthscale 4 and variance 0.01. Those hyperparameter choices are selected based on estimations using ground truth observations.

The considered spatial region is $21 \times 21$ with (mildly) uneven grid, and the time horizon is $[0, 5]$ with time step 0.01. A uniformly drawn initial observer is placed, followed by 9 additional observers using the different policies. A hundred runs with different initial seeds are conducted, with their performance measured in the same way as before.

The result in Figure 7 indicates that BALLAST-true and BALLAST-opt consistently outperform other policies with noticeable margins. At the end of the deployment, the two BALLAST policies save around 2 drifters against the benchmark, which yields about $22\%$ cost saving. We also notice DIST-SEP performing worse than EIG both here and above. As this heuristic was proposed as an approximation to information gain (see Section 4.2 of Chen et al. (2024b)), we find DIST-SEP's underperformance unsurprising.

# 6. Conclusion

We apply active learning to the Lagrangian observer deployment for spatio-temporal vector fields. After noticing the inadequacy of standard methods, we introduce BALLAST to sample hypothesised vector fields and simulate potential observation trajectories for better utility measurement. We also propose VaSE to speed up computation, which could be of independent interest. Finally, our numerical experiments provide promising results on our method's effectiveness.

**General Applicability**   While BALLAST is motivated by active learning with drifters, the proposed method can be extended to other scenarios. The deployment of collar sensors for animal movement (Handcock et al., 2009) and balloon sensors for meteorological data (Wang et al., 2020) are examples where a BALLAST-style policy can be applied. Our proposed method also contributes to the growing literature of sequential designs with search constraints (Folch et al., 2024; Mutny et al., 2023; Qing et al., 2025).

**Possible Extensions**   One extension is to employ other surrogate models. Within the Gaussian process model class, there exist other physics-informed models (Hamelijnck et al., 2021; 2024; Xu & Pan, 2024). Additionally, one could also consider deep adaptive designs (Foster et al., 2021; Huang et al., 2024; Iqbal et al., 2024) to amortise the acquisition optimisation for faster decisions at deployment.

# Impact Statement

This work presents an algorithm to optimise the placement of oceanographic drifters, with the primary aim of improving environmental monitoring of ocean dynamics. Enhanced observation of currents can contribute to more accurate climate models, a better understanding of marine ecosystems, and improved responses to environmental hazards such as oil spills or extreme weather events. Meanwhile, we acknowledge that similar methodologies could also be applied in industrial contexts, including oil and gas exploration and operations.

# Acknowledgements

RZ is supported by EPSRC-funded STOR-i Center for Doctoral Training (grant no. EP/S022252/1). RZ, LA and EC are supported by the ARC ITRH for Transforming energy Infrastructure through Digital Engineering (TIDE), Grant No. IH200100009. RZ thanks Ben Lowery for the help with accessing the compute cluster, as well as William Laplante, Adrien Corenflos, and Matthias Sachs for discussions on SPDE-GP.

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

# A. Full BALLAST Algorithm

---

**Algorithm 2** BALLAST-EIG Active Learning of Lagrangian Observers

---

1: **Input:** Spatial grid $R$. Terminal time $T$. Stepsize $\delta_t$. Temporal Helmholtz GP $f$ with kernel hyperparameter $\theta$ and its extension $\boldsymbol{f} = [f, \partial_t f]^T$. Deployment number $M$. BALLAST sample number $J$. ODE solver of choice (e.g. Euler's method).

2: Initialise a Lagrangian observer randomly in $R$ at time $t_0 = 0$.

3: **for** $m = 1, 2, \ldots, M$ **do**

4:     Denote collected observations as $\mathcal{D}_m = \{X_m, y_m\}$ and set the time as $t_m = t_{m-1} + \delta_t$.

5:     Estimate kernel hyperparameter $\theta$ of $\boldsymbol{f}$ and observation noise $\sigma_{\text{obs}}^2$ using $\mathcal{D}_m$.

6:     Obtain the posterior predictive distribution $\boldsymbol{f}|\mathcal{D}_m$ marginal on $R \times t_m$.

7:     **for** $j = 1, 2, \ldots, J$ *(parallelizable)* **do**

8:         Sample $\boldsymbol{f}^{(j)}(R, t_m)$ from marginal posterior predictive distribution.

9:         Propagate $\boldsymbol{f}^{(j)}(R, t_m)$ using the SPDE approach over $[t_m, T]$ at stepsize $\delta_t$ to obtain sampled vector fields $F^{(j)}$.

10:         **for** $\boldsymbol{s} \in R$ *(parallelizable)* **do**

11:             Simulate the trajectory of a placed Lagrangian observer at $(\boldsymbol{s}, t_m)$ in vector field $F^{(j)}$ using the ODE solver to obtain $P^T(\boldsymbol{s}, t_m, F^{(j)})$.

12:             Simulate the trajectory of existing Lagrangian observers in vector field $F^{(j)}$ using the ODE solver to obtain $P^T(\boldsymbol{s}_{\text{existing}}, t_m, F^{(j)})$.

13:             Combine the trajectories into $P_j^T(\boldsymbol{s}_{\text{ag}}) = P^T(\boldsymbol{s}, t_m, F^{(j)}) \cup P^T(\boldsymbol{s}_{\text{existing}}, t_m, F^{(j)})$.

14:             Compute the utility contribution from $P_j^T(\boldsymbol{s})$ via

$$\log\det\left(I + \sigma_{\text{obs}}^2 K\left(X_m^{+P_j^T(\boldsymbol{s}_{\text{ag}})}\right)\right).$$

15:         **end for**

16:     **end for**

17:     Aggregate the utility contributions from the $J$ sampled vector fields and apply the acquisition

$$\boldsymbol{s}_m^* = \text{argmax}_{\boldsymbol{s} \in R} \frac{1}{J} \sum_{j=1}^{J}$$
$$\left[\log\det\left(I + \sigma_{\text{obs}}^2 K\left(X_m^{+P_j^T(\boldsymbol{s}_{\text{ag}})}\right)\right)\right].$$

18:     Initialise an additional Lagrangian observer at $\boldsymbol{s}_m^*$.

19: **end for**

---

# B. Mathematical Backgrounds

### B.1. Gaussian Process

A Gaussian process (GP) $f \sim \mathcal{GP}(\mu, k_\theta)$ is a stochastic process with mean function $\mu$ and kernel $k_\theta$ of hyperparameter $\theta$. Let the input space be $\mathbb{R}^m$, and the output space be $\mathbb{R}$. For notational simplicity, we also set the mean function to be zero.

A Gaussian process marginal on a finite set of test locations $x_* \in \mathbb{R}^{N_{\text{test}}}$, denoted by $f(x_*)$, is by definition a multivariate Gaussian with mean vector $\mu(x_*) \in \mathbb{R}^{N_{\text{test}}}$ and covariance Gram matrix $K_{**} := K_\theta(x_*, x_*) \in \mathbb{R}^{N_{\text{test}} \times N_{\text{test}}}$. To obtain a sample $f^{(1)}$ from such a marginal distribution, we would have

$$f^{(1)}(x_*) = \mu(x_*) + \sqrt{K_{**}}\,\xi, \qquad \xi \sim N(0, I_{N_{\text{test}}})$$

where $\sqrt{K_{**}}$ is the matrix square root of $K_{**}$, which could be obtained using multiple methods (e.g. eigendecomposition, Cholesky decomposition) (Trefethen & Bau III, 1997).

Assuming we make noisy observations $\mathcal{D} = \{(x_i, y_i)\}_{i=1}^{N_{\text{obs}}}$ such that, for any $i = 1, 2, \ldots, N_{\text{obs}}$,

$$y_i = f(x_i) + \varepsilon_i, \qquad \varepsilon_i \sim N(0, \sigma_{\text{obs}}^2).$$

The log likelihood function $l$ with parameter $\beta = (\theta, \sigma_{\text{obs}})$ for observations $\mathcal{D}$ is given by

$$l(\beta|\mathcal{D}) = -\frac{1}{2}\overline{y}^T (K_\theta(X, X) + \sigma_{\text{obs}}^2 I)^{-1} \overline{y}$$
$$- \frac{1}{2}\log\left|K_\theta(X, X) + \sigma_{\text{obs}}^2 I\right|$$
$$- N_{\text{obs}}\log 2\pi$$

where $X = [\boldsymbol{x}_1, \boldsymbol{x}_2, \ldots, \boldsymbol{x}_{N_{\text{obs}}}]^T \in \mathbb{R}^{N_{\text{obs}} \times m}$, $\overline{y} = [y_1, y_2, \ldots, y_{N_{\text{obs}}}]^T \in \mathbb{R}^{N_{\text{obs}}}$, and $K_\theta(X, X)$ denote the Gram matrix of kernel $k_\theta$ between input $X$ and $X$.

Conditional on the observations $\mathcal{D} = \{(X, y)\}$, the posterior predictive distribution at test points $x_* \in \mathbb{R}^{N_{\text{test}} \times m}$ is given by

$$f(x_*)|\mathcal{D} \sim N(\mu_*, \Sigma_*)$$
$$\mu_* = K_*^T(K + \sigma_{\text{obs}}^2 I)^{-1} y$$
$$\Sigma_* = K_{**} - K_*^T(K + \sigma_{\text{obs}}^2 I)^{-1} K_*$$

where we have the Gram matrices

$$K_{**} = K_\theta(x_*, x_*) \in \mathbb{R}^{N_{\text{test}} \times N_{\text{test}}},$$
$$K_* = K_\theta(X, x_*) \in \mathbb{R}^{N_{\text{obs}} \times N_{\text{test}}},$$
$$K = K_\theta(X, X) \in \mathbb{R}^{N_{\text{obs}} \times N_{\text{obs}}}.$$

Using the vanilla GP formulation presented above, the computational cost of likelihood training is $O(N_{\text{obs}}^3)$, while the cost of prediction at $N_{\text{test}}$ test points is $O(N_{\text{obs}}^3 + N_{\text{test}}^3 + N_{\text{obs}}^2 N_{\text{test}} + N_{\text{obs}} N_{\text{test}}^2)$.

## B.2. Helmholtz GP

The Helmholtz GP of Berlinghieri et al. (2023) is a vector-valued (Alvarez et al., 2012) GP. For a vector field $F$, the Helmholtz decomposition (Bhatia et al., 2012) breaks it down as the linear combination of the potential function $\Phi$ and stream function $\Psi$ as

$$F = \text{grad}\,\Phi + \text{rot}\,\Psi$$

for differential operators grad and rot. By imposing a GP structure to the potential and stream functions, i.e. $\Phi \sim \mathcal{GP}(0, k_\Phi)$ and $\Psi \sim \mathcal{GP}(0, k_\Psi)$, we have the Helmholtz kernel $F \sim \mathcal{GP}(0, k_{\text{Helm}})$ using the property of kernel under linear operators (Agrell, 2019)

$$k_{\text{Helm}}(\boldsymbol{x}, \boldsymbol{x}') =$$
$$\begin{bmatrix} \partial^2_{x_1 x_1'} k_\Phi + \partial^2_{x_2 x_2'} k_\Psi & \partial^2_{x_1 x_2'} k_\Phi - \partial^2_{x_2 x_1'} k_\Psi \\ \partial^2_{x_2 x_1'} k_\Phi - \partial^2_{x_1 x_2'} k_\Psi & \partial^2_{x_2 x_2'} k_\Phi + \partial^2_{x_1 x_1'} k_\Psi \end{bmatrix}$$

for $\boldsymbol{x}, \boldsymbol{x}' \in \mathbb{R}^2$ if we assume $\Phi$ and $\Psi$ are independent. The Helmholtz kernel with dependent potential and stream function can be similarly obtained using linear properties of the kernel – see Section 2.1 of Ponte et al. (2024) for the kernel expression.

## B.3. Information Theory

Consider a continuous random variable $X$ with probability density function $p(x)$. Its (differential) **entropy** $H(X)$ is provided by Cover & Thomas (2006)

$$H(X) := \mathbb{E}_{x \sim X}[-\log p(x)] = \int -p(x) \log p(x) dx.$$

For example, the entropy of a multivariate Gaussian $H(X)$ where $X \sim N_d(\mu, \Sigma)$ is given by

$$H(X)$$
$$= -\mathbb{E}_{x \sim X}[\log p(x)]$$
$$= \mathbb{E}\left[\frac{d}{2}\log \pi + \frac{1}{2}\log \det \Sigma + \frac{1}{2}(x - \mu)^T \Sigma^{-1}(x - \mu)\right]$$
$$= \frac{d}{2}\log \pi + \frac{1}{2}\log \det \Sigma + \frac{1}{2}\text{tr}\left[\Sigma^{-1} \mathbb{E}\left[(x - \mu)^T (x - \mu)\right]\right]$$
$$= \frac{d}{2}\log \pi + \frac{1}{2}\log \det \Sigma + \frac{1}{2}\text{tr}\left[\Sigma^{-1}\Sigma\right]$$
$$= \frac{d}{2}\log \pi + \frac{d}{2} + \frac{1}{2}\log \det \Sigma.$$

For two continuous random variables $X, Y$ with joint density $p(x, y)$ and individual densities $p_X, p_Y$ respectively, the **joint entropy** of $X, Y$ is defined as

$$H(X, Y) := \mathbb{E}_{(x,y) \sim (X,Y)}[-\log p(x, y)]$$
$$= \iint -p(x, y) \log p(x, y) dx dy.$$

The **conditional entropy** of $X$ given $Y$ is defined as

$$H(X|Y) := \mathbb{E}_{(x,y) \sim (X,Y)}[-\log p(x|y)]$$
$$= \iint -p(x, y) \log \frac{p(x, y)}{p(y)} dx dy.$$

When $X, Y$ are independent, so $p(x, y) = p_X(x) p_Y(y)$ for any $x, y$, we have the identity

$$H(X, Y) = H(X) + H(Y), \qquad H(X|Y) = H(X).$$

Subsequently, we define the **mutual information** between $X$ and $Y$ as the measure of mutual dependency between the two random variables, calculated as

$$I(X; Y) := H(X) + H(Y) - H(X, Y)$$
$$= H(X) - H(X|Y) = H(Y) - H(Y|X)$$

which can also be viewed as the Kullback-Leibler divergence between the density of the joint distribution $p(x, y)$ and the outer product distribution $p(x) \otimes p(y)$.

# C. Expected Information Gain Computation for Gaussian Process Surrogates

Following Section B.1, a GP $f \sim \mathcal{GP}(0, k)$ and noisy observations $\mathcal{D} = \{(x_i, y_i)\}_{i=1}^{N_{\text{obs}}}$ with i.i.d. Gaussian noises $y_i = f(x_i) + \varepsilon_i$ for $\varepsilon_i \sim N(0, \sigma_{\text{obs}}^2 I)$. The posterior predictive distribution at $N_{\text{test}}$ test points $x_*$ is a multivariate Gaussian distribution by Gaussian process conjugacy, i.e.

$$f(x_*)|\mathcal{D} \sim N(\mu_*, \Sigma_*)$$
$$\mu_* = K_*^T (K + \sigma_{\text{obs}}^2 I)^{-1} y$$
$$\Sigma_* = K_{**} - K_*^T (K + \sigma_{\text{obs}}^2 I)^{-1} K_*.$$

Following the result of Section B.3, the entropy of a multivariate Gaussian is linked to the log determinant of its covariance matrix. So, the entropy of the posterior predictive at finitely many test points is given by

$$H(f(x_*)|\mathcal{D})$$
$$= \frac{1}{2}\log \det \Sigma_* + \text{const}$$
$$= \frac{1}{2}\log \det \left(K_{**} - K_*^T (K + \sigma_{\text{obs}}^2 I)^{-1} K_*\right) + \text{const}.$$

For a hypothetical observation location $x$ and its measurement $y$, the information gain of observing this additional fictitious observation is provided by the difference between posteriors $f|\mathcal{D}$ and $f|\mathcal{D} \cup \{(x, y)\}$, so

$$IG(y) = H(f|\mathcal{D}) - H(f|\mathcal{D} \cup \{(x, y)\}).$$

For finite test points $x_*$, we can further simplify the above

expression to

$$IG(y) = H(f|\mathcal{D}) - H(f|\mathcal{D} \cup \{(x,y)\})$$
$$= \log \det \Sigma_* - \log \det \Sigma_*^+,$$
$$\Sigma_* = K_{**} - K_*^T(K + \sigma_{\text{obs}}^2 I)^{-1}K_*,$$
$$\Sigma_*^+ = K_{**} - (K_*^+)^T(K^+ + \sigma_{\text{obs}}^2 I^+)^{-1}K_*^+,$$
$$X^+ = X \cup x,$$
$$\mathcal{D}^+ = \mathcal{D} \cup \{(x,y)\},$$
$$K^+ = K(X^+, X^+),$$
$$K_*^+ = K(X^+, x_*).$$

Furthermore, we notice that there is no dependency of the observation value $y$ in the above expression of information gain, which means the expected information gain is identical to the information gain, i.e.

$$EIG(x) = \mathbb{E}_{g(y|x)}[IG(y)] = \log \det \Sigma_* - \log \det \Sigma_*^+.$$

Although a closed-form expression for the EIG acquisition function exists in active learning with Gaussian process surrogates under Gaussian observation noises, the computation cost of the above formulation is still high, involving calculating the posterior predictive covariance matrix and its determinant. In particular, for each possible measurement point $x$, the computational cost of calculating $EIG(x)$ is $O(N_{\text{test}}^3 + N_{\text{obs}}^3 + N_{\text{obs}}^2 N_{\text{test}} + N_{\text{obs}} N_{\text{test}}^2)$.

### C.1. Reformulation of Expected Information Gain

Fortunately, we can reformulate the EIG to greatly reduce the computational costs using the property of mutual information (see Section B.3 for definitions). The expression presented below appears in Section 2.2 of Srinivas et al. (2010) too.

Instead of focusing on the marginal distribution of the GP at finitely many test locations, we consider the full distribution $p(f)$ and look at its expected information gain for additional observations. For a GP $p(f)$ and observations $\mathcal{D}_A = \{(x_A, y_A)\}$ with $y_A = f(x_A) + \varepsilon, \varepsilon \sim N(0, \sigma_{\text{obs}}^2 I)$, we have

$$IG(y_A)$$
$$= H(p(f)) - H(p(f|\mathcal{D}_A))$$
$$= MI(f; \mathcal{D}_A)$$
$$= H(p(\mathcal{D}_A)) - H(p(\mathcal{D}_A|f))$$

using the symmetry property of mutual information between two random variables. Since $y_A = f(x_A) + \varepsilon$, the covariance matrix of $y_A$ is $K(x_A, x_A) + \sigma_{\text{obs}}^2 I$. Also, the covari-

ance of $y_A|f$ is merely $\sigma_{\text{obs}}^2 I$. Thus, we have

$$IG(y_A)$$
$$= H(p(\mathcal{D}_A)) - H(p(\mathcal{D}_A|f))$$
$$= \frac{1}{2}\log \det(K(x_A, x_A) + \sigma_{\text{obs}}^2 I) - \frac{1}{2}\log \det(\sigma_{\text{obs}}^2 I)$$
$$= \frac{1}{2}\log \det(\sigma_{\text{obs}}^{-2} K(x_A, x_A) + I).$$

Using the above result, we consider $\mathcal{D}_B = \mathcal{D}_A \cup \{(x,y)\} = \{(x_B, y_B)\}$ the observations set with additional observation $(x,y)$ and can compute the following information gain

$$IG(y)$$
$$= H(p(f|\mathcal{D}_A)) - H(p(f|\mathcal{D}_B))$$
$$= H(p(f|\mathcal{D}_A)) - H(p(f)) + H(p(f)) - H(p(f|\mathcal{D}_B))$$
$$= -\frac{1}{2}\log \det(\sigma_{\text{obs}}^{-2} K(x_A, x_A) + I)$$
$$\quad + \frac{1}{2}\log \det(\sigma_{\text{obs}}^{-2} K(x_B, x_B) + I)$$

and therefore

$$\text{argmax}_x \, EIG(x)$$
$$= \text{argmax}_x \, \mathbb{E}_{y \sim p(y|\mathcal{D}_A, x)}[IG(y)]$$
$$= \text{argmax}_x \left[ -\frac{1}{2}\log \det(\sigma_{\text{obs}}^{-2} K(x_A, x_A) + I) \right.$$
$$\quad \left. + \frac{1}{2}\log \det(\sigma_{\text{obs}}^{-2} K(x_B, x_B) + I) \right]$$
$$= \text{argmax}_x \log \det(\sigma_{\text{obs}}^{-2} K(x_B, x_B) + I).$$

This reformulation of the expected information gain is computationally cheap, and the computation for $EIG(x)$ for any $x$ is merely $O(N_{\text{obs}}^3)$.

## D. Proof of Proposition 1

Here, we formalise Proposition 1 and provide a proof.

**Proposition 2** (Formalisation of Prop 1). *Consider the sequential experimental design problem with existing observation $\mathcal{D}$, measurement set $X$, and utility $U$ where the observations are made by Lagrangian observers (see Section I.1 for details). At any decision time $t$, the deployment position $x^S$ following standard utility is suboptimal w.r.t. to the true utility considering all potential observations made by the placed observer.*

*Proof.* At any decision time $t$, the standard utility construction that only considers the initial placement location, i.e.

$$x^S := \text{argmax}_{x \in X} \, \mathbb{E}_{p(y|\mathcal{D}, x)}[U(y)]$$

while the Lagrangian utility $LU$ accounting for all potential observations made by the placed observer yields the decision $x^*$ defined as

$$x^* := \text{argmax}_{x \in X} LU(x)$$
$$:= \text{argmax}_{x \in X} \mathbb{E}\left[U\left(\int_t^T y_s ds\right)\right]$$

where $T$ is the terminal time of the experimental design. The decision from standard utility construction is suboptimal, in the sense that

$$LU(x^S) \leq LU(x^*).$$

This follows directly from the definition, as $x^*$ is constructed to be the maximiser of $LU$, any other value $x \in X$ will not produce $LU(x)$ that is greater than $LU(x^*)$. Since $x^S \in X$, the desired inequality $LU(x^S) \leq LU(x^*)$ holds. $\qquad \square$

We should remark that the BALLAST utility of (4) approximates the Lagrangian utility $LU$ above, where the integral is replaced by the sum of discretised observer trajectories.

## E. Computational Tricks

### E.1. Kronecker Products

Given two matrices $A \in \mathbb{R}^{m \times n}, B \in \mathbb{R}^{p \times q}$, the **Kronecker product** $A \otimes B$ is defined as

$$A \otimes B = \begin{bmatrix} a_{11}B & \cdots & a_{1n}B \\ \vdots & \ddots & \vdots \\ a_{m1}B & \cdots & a_{mn}B \end{bmatrix}.$$

Below, we will state several key properties of Kronecker products and establish a computationally efficient Kronecker matrix-vector product. The basic properties of the Kronecker product can be established from the definition, and additional details can be found in Chapter 5.2 of Saatçi (2012).

For matrices $A, B, C, D$ with suitable sizes such that the following operations make sense, we have

- $(A \otimes B)(C \otimes D) = (AC) \otimes (BD)$

- $(A \otimes B)^T = A^T \otimes B^T.$

Note that a direct consequence of the above properties is that, for matrices admitting Cholesky decomposition $P = L_P L_P^T$ and $Q = L_Q L_Q^T$, we have

$$P \otimes Q = (L_P L_P^T) \otimes (L_Q L_Q^T)$$
$$= (L_P \otimes L_Q)(L_P^T \otimes L_Q^T)$$
$$= (L_P \otimes L_Q)(L_P \otimes L_Q)^T$$

and thus the lower triangular matrix for the Cholesky decomposition of $P \otimes Q$ is given by $L_P \otimes L_Q$.

Before stating the matrix-vector product result, we first need to define the **vectorization** operation. For a matrix $A \in \mathbb{R}^{m,n}$, its vectorization $\text{vec}(A)$ is a column vector that concatenates the column vectors of $A$ from left to right, i.e.

$$\text{vec}(A) = [a_{11}, \ldots, a_{m1}, \ldots, a_{1n}, \ldots, a_{mn}]^T.$$

**Proposition 3.** *For matrix $A \in \mathbb{R}^{m \times n}, B \in \mathbb{R}^{p \times q}, X \in \mathbb{R}^{n \times p}$, we have*

$$\text{vec}(AXB) = (B^T \otimes A)\text{vec}(X).$$

*Proof.* First, we consider the $k$-th column of the matrix product $AXB$, which can be expressed as below,

$$(AXB)_{:,k} = ((AX)B)_{:,k} = (AX)B_{:,k} = A(XB_{:,k})$$
$$= A\sum_{i=1}^p X_{:,i} B_{i,k} = \sum_{i=1}^p B_{i,k} AX_{:,i}$$
$$= \begin{bmatrix} B_{1,k}A & B_{2,k}A & \cdots & B_{p,k}A \end{bmatrix}\text{vec}(X)$$
$$= (B_{:,k}^T \otimes A)\text{vec}(X).$$

Next, using the above expression, the vectorization $\text{vec}(AXB)$ is a vertical stack of the above quantity, so we have

$$\text{vec}(AXB) = \begin{bmatrix} (AXB)_{:,1} \\ (AXB)_{:,2} \\ \vdots \\ (AXB)_{:,q} \end{bmatrix}$$
$$= \begin{bmatrix} (B_{:,1}^T \otimes A)\text{vec}(X) \\ (B_{:,2}^T \otimes A)\text{vec}(X) \\ \vdots \\ (B_{:,q}^T \otimes A)\text{vec}(X) \end{bmatrix}$$
$$= \begin{bmatrix} B^T \otimes A \end{bmatrix}\text{vec}(X).$$

$\qquad \square$

It can be observed immediately that the left-hand-side expression of the quantity $\text{vec}(AXB)$ uses less storage and computes faster than the right-hand-side expression with Kronecker product $B^T \otimes A$.

### E.2. Rank-q Gram Matrix Updates

For a kernel $k$, we denote the Gram matrix generated under this kernel at inputs $X, Y$ as $K(X, Y)$ such that $K(X, Y)_{i,j} = k(X_i, Y_j)$, and denote $K(X, X) = K(X)$ for simplicity. With Gaussian processes, we may consider computations with $K(X \cup X_*)$ when we have already computed $K(X)$ at an earlier time. For $X_*$ of size $q$, such computations are often denoted as the rank-$q$ updates of Gram

matrices, and the updated Gram matrix is of the following form

$$K(X \cup X_*) = \begin{bmatrix} K(X) & K(X, X_*) \\ K(X_*, X) & K(X_*) \end{bmatrix}$$

with $K(X, X_*) = K(X_*, X)^T$.

Here, we will describe how we can compute the determinant more efficiently with rank-$q$ updates, as such computations are repeatedly conducted for the utility computation, such as (5). This relies on the following result of the block matrix determinant.

**Proposition 4.** *For invertible matrix A, we have*

$$\det \begin{bmatrix} A & B \\ C & D \end{bmatrix} = \det(A) \det(D - CA^{-1}B).$$

Therefore, to efficiently compute the determinant of the Gram matrices $K(X \cup X_*)$ with fixed $X$ and different $X_*$, we could first compute the lower Cholesky decomposition for $K(X) = LL^T$, which gives us the determinant and inverse as

$$\det K(X) = \left( \prod_i L_{ii} \right)^2, \qquad K(X)^{-1} = L^{-T}L^{-1}$$

and thus we have

$$\begin{aligned}
&\det K(X \cup X_*) \\
&= \det K(X) \det \Big( K(X_*) \\
&\qquad - K(X, X_*)^T L^{-T} L^{-1} K(X, X_*) \Big) \\
&= \det K(X) \det \Big( K(X_*) \\
&\qquad - [L^{-1}K(X, X_*)]^T [L^{-1}K(X, X_*)] \Big).
\end{aligned}$$

Similar reformulations can be applied for the determinant computation of (5). Such a rank-$q$ update would be used as the default for the computation in this work.

# F. The SPDE Approach to Gaussian Process Regression

Consider a spatio-temporal GP $f(x) \sim \mathcal{GP}(0, k)$ with $x = (\boldsymbol{s}, t) \in \mathbb{R}^3$, $\boldsymbol{s} \in \mathbb{R}^2$, $t \in \mathbb{R}$ and separable kernel $k(x, x') = k_{\text{space}}(\boldsymbol{s}, \boldsymbol{s}') k_{\text{time}}(t, t')$ where temporal kernel $k_{\text{time}}$ is set to be Matérn-$3/2$. Below, we will describe the details of the dynamic formulation of such a GP using the stochastic partial differential equation (SPDE) approach (Solin, 2016).

## F.1. State-Space Formulation of the Temporal Component

First, consider a zero-mean temporal GP with a Matérn $\frac{3}{2}$ kernel in isolation. Let $l$ denote the kernel's length-scale and

$\sigma^2$ denote its variance. We would also define $\lambda := \sqrt{3}/l$ for simplicity. This process $\{h(t)\}_t$ can be modelled as the solution to a stochastic differential equation (SDE). In companion (state-space) form, the temporal dynamics are given by

$$\begin{aligned}
\frac{d}{dt}\boldsymbol{h}(t) &= \frac{d}{dt} \begin{bmatrix} h(t) \\ \frac{d}{dt}h(t) \end{bmatrix} \\
&= \underbrace{\begin{bmatrix} 0 & 1 \\ -\lambda^2 & -2\lambda \end{bmatrix}}_{F} \begin{bmatrix} h(t) \\ \frac{d}{dt}h(t) \end{bmatrix} + \underbrace{\begin{bmatrix} 0 \\ 1 \end{bmatrix}}_{L} w(t),
\end{aligned}$$

driven by the white noise process $w(t)$ with spectral density matrix $Q_c = 4\lambda^3\sigma^2 I_2$. For this SDE, the *exact* one-step transition with stepsize $\delta_t$ is given by

$$\boldsymbol{h}(t + \delta_t) = \Phi\boldsymbol{h}(t) + \xi, \qquad \xi \sim N(0, Q)$$

where

$$\begin{aligned}
\Phi &= \exp\left( F\,\delta_t \right), \\
Q &= P_\infty - \Phi\,P_\infty\,\Phi^T, \\
P_\infty &= \begin{bmatrix} \sigma^2 & 0 \\ 0 & \lambda^2\sigma^2 \end{bmatrix}
\end{aligned}$$

and $P_\infty$ is the covariance matrix for the equilibrium distribution of the SDE.

## F.2. State-Space Formulation of the spatio-temporal Model

Assume the spatial grid we are interested in is denoted by $R$ with $N_{\text{space}}$ points. The corresponding spatial Gram matrix with kernel $k_{\text{space}}$ is denoted by $K_{\text{space}} \in \mathbb{R}^{DN_{\text{space}} \times DN_{\text{space}}}$ where $D$ is the output dimension.

For a single spatial location $\boldsymbol{s}$ and time $t$, the extended state is $\mathbf{f}(\boldsymbol{s}, t) = \begin{bmatrix} f(\boldsymbol{s}, t) & \partial_t f(\boldsymbol{s}, t) \end{bmatrix}^T$. Because the spatial and temporal components are separable by construction, we can incorporate the spatial dimensions into the evolution using Kronecker products $\otimes$, giving us the SPDE

$$\frac{d}{dt}\boldsymbol{f}(R, t) = (I_{\text{space}} \otimes F)\boldsymbol{f}(R, t) + (I_{\text{space}} \otimes L)\boldsymbol{w}(t)$$

driven by the white noise process $\boldsymbol{w}(t)$ with spectral density matrix $Q_{\text{full}} = K_{\text{space}} \otimes Q_c$. Here, $I_{\text{space}}$ is the $DN_{\text{space}} \times DN_{\text{space}}$ identity matrix. Subsequently, the one-step transition from $f_k$ to $f_{k+1}$ with stepsize $\delta_t$ is given by

$$f_{k+1} = \Phi_{\text{full}}f_k + e_k, \quad e_k \sim N\left(0, Q_{\text{full}}\right),$$

where $\Phi_{\text{full}} = I_{\text{space}} \otimes \Phi$ and $Q_{\text{full}} = K_{\text{space}} \otimes Q$. The inclusion of a spatial component at each time changes the white noise process driving the SDE and turns the full equation into an SPDE. Like with the temporal GP case, this is a mere reformulation, and no approximation happened.

We can extract the GP of interest $f$ from the full state vector $\mathbf{f}(R,t) = \begin{bmatrix} f(R,t) & \partial_t f(R,t) \end{bmatrix}^T$ using the measurement operator $H_{\text{full}}$ defined by

$$H_{\text{full}} = I_{\text{space}} \otimes \begin{bmatrix} 1 & 0 \end{bmatrix},$$

so that the GP of interest is extracted via $f(R,t) = H_{\text{full}}\mathbf{f}(R,t)$.

At time $t_k$, when we make observations at a subset of the full spatial grid $R$, we could construct a measurement operator $H_k$ that selects the right coordinates of the full state, i.e. we would have

$$\boldsymbol{y}_k = H_k\boldsymbol{f}(R,t_k) + \boldsymbol{\varepsilon}_k, \qquad \boldsymbol{\varepsilon}_k \sim N(0, \sigma_{\text{obs}}^2 I)$$

Therefore, the state-space formulation of the spatio-temporal GP of interest is given by

$$\begin{aligned} f_{k+1} &= \Phi_{\text{full}}f_k + \xi_k, \qquad \xi_k \sim N\left(0, Q_{\text{full}}\right), \\ y_k &= H_k f_k + \varepsilon_k, \qquad \varepsilon_k \sim N(0, \sigma_{\text{obs}}^2 I). \end{aligned}$$

for observation time indices $k = 0, 1, \ldots, T$.

## F.3. Regression as Sequential Inference

The state-space formulation of spatio-temporal GP allows us to consider the GP dynamically and enables the regression task to be converted to a filtering and smoothing task. In particular, as we know the exact, analytical transition and emission dynamics of the state space model, we can apply a Kalman filter and a Rauch-Tung-Striebel (RTS) smoother (Särkkä & Solin, 2019).

GP regression is equivalent to doing the filtering and then smoothing of the observations. For posterior prediction, if the prediction time is after the last observation time, one would use the state-space model transition formula; if the prediction time is before the last observation time but different from any observation time, one would include it in the filtering step, then be smoothed. Prediction at a new location requires re-running the filtering and smoothing by extending the new location into the spatial grid $R$.

Below, we will present the filtering and smoothing at a regular time grid indexed $k = 0, 1, \ldots, T$ where the observation times are a subset of it. Also, the subscript $h|j$ of mean $m$ and covariance $P$ represents the mean and covariance at time index $h$ conditional on the observations until time index $j$.

### F.3.1. KALMAN FILTERING

The Kalman filter proceeds by alternating between the propagation step and the assimilation step.

**Propagation Step:** From the filtered state estimate at time $k$, with mean $m_{k|k}$ and covariance $P_{k|k}$, we predict the state at time $k+1$:

$$\begin{aligned} m_{k+1|k} &= \Phi_{\text{full}}\, m_{k|k}, \\ P_{k+1|k} &= \Phi_{\text{full}}\, P_{k|k}\, \Phi_{\text{full}}^T + Q_{\text{full}}. \end{aligned}$$

We assume the SPDE begins at equilibrium with initial mean $m_{0|0} = 0 \in \mathbb{R}^{2DN_{\text{space}}}$ and initial covariance $P_{0|0} = K_{\text{space}} \otimes P_\infty$.

**Assimilation Step:** When an observation $y_{k+1}^{\text{obs}}$ is available, we define a time-dependent observation matrix $H_{k+1}$ selecting the observed locations and perform the update:

$$\begin{aligned} v_{k+1} &= y_{k+1}^{\text{obs}} - H_{k+1}\, m_{k+1|k}, \\ S_{k+1} &= H_{k+1}\, P_{k+1|k}\, H_{k+1}^T + R_{k+1}, \\ K_{k+1} &= P_{k+1|k}\, H_{k+1}^T\, S_{k+1}^{-1}, \\ m_{k+1|k+1} &= m_{k+1|k} + K_{k+1}\, v_{k+1}, \\ P_{k+1|k+1} &= P_{k+1|k} - K_{k+1}\, H_{k+1}\, P_{k+1|k}. \end{aligned}$$

### F.3.2. RTS SMOOTHING

After running the Kalman filter over the fine time grid, the RTS smoother refines the estimates using future observations. For $k = T-1, T-2, \ldots, 0$, the smoother performs:

$$\begin{aligned} J_k &= P_{k|k}\, \Phi_{\text{full}}^T\, \left(P_{k+1|k}\right)^{-1} \\ m_{k|T} &= m_{k|k} + J_k\left(m_{k+1|T} - m_{k+1|k}\right), \\ P_{k|T} &= P_{k|k} + J_k\left(P_{k+1|T} - P_{k+1|k}\right) J_k^T. \end{aligned}$$

The smoothed state estimates, $m_{k|T}$ and $P_{k|T}$, represent the posterior mean and covariance over the latent spatio-temporal field given all available observations at time index $k$.

## F.4. Computational Costs for Posterior Sampling

Consider a separable spatio-temporal GP with a Matérn $3/2$ temporal kernel and a Helmholtz spatial kernel. Let $N_s$ denote the number of spatial grids that we are doing sequential inference on. In such a setting, the size of the transition matrices $F_{\text{full}}$, $\Phi_{\text{full}}$, $Q_{\text{full}}$ would be of the size $2N_s \times 2N_s$ as they are all Kronecker products of $2 \times 2$ base matrices and the $N_s \times N_s$ spatial Gram matrix $K_{\text{space}}$.

To sample from such a GP model without any observation for $N_t$ time steps would involve propagating an initial condition $f_0$ using

$$\begin{aligned} f_{k+1} &= \Phi_{\text{full}}f_k + \sqrt{Q_{\text{full}}}z_k, \qquad z_k \sim N\left(0, I\right), \\ f_0 &\sim N(0, K_{\text{space}} \otimes P_\infty) \end{aligned}$$

for $k = 0, 1, \ldots, N_t$. Therefore, given the specifications of the transition, the total computational costs of prior sampling are $O((2N_s)^2 N_t)$. This can be further improved using

the Kronecker matrix-vector product described in Section E.1.

Given $N_{\text{obs}}$ observations taken at $N_{\text{obs-time}}$ observation times and $N_{\text{obs-loc}}$ observation locations, the regression of these data using the SPDE approach involves, minimally, filtering the data at $N_{\text{obs-time}}$ observation times. Each observation time requires propagation and assimilation with the costs, so the total computational cost is $O(N_{\text{obs-time}} N_{\text{obs-loc}}^3)$. Similarly, the likelihood training of these data will cost $O(N_{\text{obs-time}} N_{\text{obs-loc}}^3)$.

If one wishes to learn about the posterior predictive distribution, the prediction test points (test locations and test times) should be added to the filtering and smoothing step. For example, to predict at $N_{\text{space}}$ locations (almost completely) distinct from the observation locations, the computational cost of obtaining such a predictive distribution is

$$O((N_{\text{obs-loc}} + N_{\text{space}})^2 N_{\text{obs-time}}).$$

Subsequently, the computational cost of sampling from these posterior predictive distributions for $N_{\text{sampleT}}$ prediction times at $N_{\text{space}}$ prediction locations would be of $O((2N_{\text{space}})^2 N_{\text{sampleT}})$.

### F.5. Connection to Spatial SPDE-GP

The SPDE-GP framework we employ in this paper follows the work of Hartikainen & Särkkä (2010) and Sarkka et al. (2013), while another seemingly distinct version of Lindgren et al. (2011) and Lindgren et al. (2022) exists in the spatial statistics literature. Here, we will briefly highlight their connection and their shared origin in Whittle (1954) and Whittle (1963). In this section, we will denote the version by Hartikainen & Särkkä (2010) as the *temporal version* and the version by Lindgren et al. (2011) the *spatial version*.

Both the spatial and temporal versions are established on the following S(P)DE interpretation of the Matérn GP due to Peter Whittle (Whittle, 1954; 1963): A $d$ dimensional Matérn GP with scale parameter $\kappa$ and smoothness parameter $\nu$ is the solution to the following SPDE

$$(\kappa^2 - \Delta)^{\alpha/2} x(u) = W(u)$$

where $\Delta$ is the Laplacian, $\alpha = \nu + d/2$, $x$ is the process of interest, and $W$ is a $d$-dimensional white noise process with unit variance. The solution model has kernel variance $\sigma^2$ with

$$\sigma^2 = \frac{\Gamma(\nu)}{\Gamma(\nu + d/2)(4\pi)^{d/2} \kappa^{2\nu}}$$

where $\Gamma$ is the Gamma function, and the kernel variance can be adjusted by scaling the white noise process $W$.

The temporal version sets $d = 1$, and the spatial version sets $d = 2$. The temporal version, additionally, manipulates

the SDE into the following form, which enables a Gaussian linear model expression for Kalman filtering and RTS smoothing:

$$(\kappa + \Delta)^2 x(u) = \tilde{W}(u)$$

for an adjusted white noise process $\tilde{W}$. The above SDE is linear and admits closed-form transition densities, so its sequential inference is exact.

The spatial version of Lindgren et al. (2011) involves approximation. The method first constructs a finite-dimensional basis expansion of $x$, inspired by the finite element method, like

$$x(u) = \sum_{k=1}^{n} \phi_k(u) w_k$$

for basis function $\{\phi_k\}$ and (zero-mean) Gaussian weights $\{w_k\}$, then solve for the precision matrix for the joint Gaussian weights. This turns the original Gaussian field (alternative name for 2D spatial GP) of $x$ into a Gaussian Markov random field (Rue & Held, 2005) of $w$, which can be solved more efficiently.

## G. Vanilla SPDE Exchange Details

Let $f$ be a spatio-temporal GP with separable kernel $k = k_s k_t$ where $k_s$ is the spatial kernel and the time kernel $k_t$ is Matérn (with smoothness $3/2$ here, but can be extended to any half-integer smoothness). We are interested in a spatial grid $R$ and a temporal grid $\mathcal{T}$ that is a regular interval of timestamps in $[0, T]$ with time gap $\delta$. The temporal grid does not have to be regular, but for the simplicity of presentation, this is assumed. A non-regular temporal grid would only require us to keep track of the time increments and adjust the SPDE-GP updates accordingly, which do not affect the general vanilla SPDE exchange (VaSE) method.

Consider at time $t_n \in \mathcal{T}$ we have observations $\mathcal{D}$. The VaSE method to sample from the posterior $f | \mathcal{D}$ at full spatio-temporal grid for time $t_n$ onwards consists of three parts:

1. posterior GP update with $\mathcal{D}$ for the extended GP $\boldsymbol{f} = [f, f']^T$ using standard GP,

2. compute the posterior predictive distribution of $\boldsymbol{f}$ at $R \times \{t_n\}$,

3. sample from the posterior predictive and propagate for the remaining timestamps using SPDE-GP.

The first and last parts apply the vanilla and the SPDE methods, while the middle part acts as the exchange to switch between the two modes of GP inferences. Details of the SPDE-GP were presented in Section F.

When the smoothness parameter of the temporal Matérn kernel becomes $(2j + 1)/2$ for some nonnegative integer $j$,

we will adjust the above procedure with an extended GP $f = [f, f', \cdots, f^{(j)}]^T$ with all $j$ derivatives. This is to ensure the SPDE-GP propagation has the necessary dimensions.

### G.1. Computational Cost Analysis and Comparison

We conduct a cost analysis here. Let $N_{\text{obs}}$ denote the number of observations, $N_{\text{obs,t}}$ denote the number of distinct observation time slices, $N_{\text{pred,s}}$ and $N_{\text{pred,t}}$ denote the number of prediction spatial and temporal points. Below, we will carefully present the computational cost of drawing a posterior sample under the standard GP, SPDE-GP, and VaSE, where the cost orders for each portion of the sampling cost are included in square brackets.

Following from Section B.1, the *standard* GP posterior sample requires computing the posterior predictive distribution's covariance $[N_{\text{obs}}^3]$ and taking its matrix square root $[N_{\text{pred,s}}^3 N_{\text{pred,t}}^3]$, as well as multiplying the Gaussian noise $[N_{\text{obs}}^2 N_{\text{pred,s}} N_{\text{pred,t}}]$ and adding the computed posterior mean $[N_{\text{obs}} N_{\text{pred,s}}^2 N_{\text{pred,t}}^2]$.

Following from Section F, the *SPDE-GP* approach to draw a posterior sample consists of first creating update matrices via Kronecker products of the temporal update matrices and the full Gram matrix of all $N_{\text{pred,s}} + N_{\text{obs}}$ considered spatial points, followed by matrix inversions for assimilating observations $[(N_{\text{obs}} + N_{\text{pred,s}})^3 N_{\text{obs,t}}]$, then using the coordinates corresponding to test points to scale Gaussian noise and propagate forward to generate the sample $[N_{\text{pred,s}}^2 N_{\text{pred,t}}]$.

Finally, the *VaSE* approach generates the mean vector and covariance matrix of the posterior predictive distribution at the current time over all spatial test points $[N_{\text{obs}}^3 + N_{\text{pred,s}}^2 N_{\text{obs}} + N_{\text{pred,s}} N_{\text{obs}}^2]$, then this spatial component is used to scale Gaussian noise and propagate forward to generate the sample $[N_{\text{pred,s}}^2 N_{\text{pred,t}}]$.

In summary, the three methods' costs to generate one posterior sample are

(**Standard**) $O(N_{\text{obs}}^3 + N_{\text{pred,s}}^3 N_{\text{pred,t}}^3 + N_{\text{obs}}^2 N_{\text{pred,s}} N_{\text{pred,t}}$
$\qquad + N_{\text{obs}} N_{\text{pred,s}}^2 N_{\text{pred,t}}^2)$,

(**SPDE**) $O((N_{\text{obs}} + N_{\text{pred,s}})^3 N_{\text{obs,t}} + N_{\text{pred,s}}^2 N_{\text{pred,t}})$,

(**VaSE**) $O(N_{\text{obs}}^3 + N_{\text{pred,s}}^2 N_{\text{obs}} + N_{\text{pred,s}} N_{\text{obs}}^2$
$\qquad + N_{\text{pred,s}}^2 N_{\text{pred,t}})$.

## H. Ablation Studies

Here, we conduct two ablation studies on the BALLAST algorithm outlined in Algorithm 1 to investigate the appropriate choice of sample number $J$ and the length of the forward projection time horizon.

### H.1. Sample Number

This study investigates the choice of posterior sample number $J$ of the BALLAST utility. In particular, we have the following acquisition function of BALLAST-EIG:

$$\mathbb{E}_F \left[ \log \det \left( I + \sigma_{\text{obs}}^2 K(X_n^{+P^T(s)}, X_n^{+P^T(s)}) \right) \right]$$

where $F$ is the random vector field following the posterior distribution, $\sigma_{\text{obs}}^2$ is the variance of the observation noise, $X_n$ is the existing observations' locations, $P^T(s)$ is the projected trajectory locations of a drifter deployed at $s$ as well as the existing drifters from the current time till time $T$ (also the terminal time of the deployment) under the random vector field $F$, and $X_n^{+P^T(s)} = X_n \cup P^T(s)$ is the aggregated observation locations.

The above quantity does not admit a closed-form expression, and we will approximate it using Monte Carlo with samples $F^{(1)}, F^{(2)}, \ldots, F^{(J)}$ from the posterior. This gives us the following:

$$B(s; J|X_n) := \frac{1}{J} \sum_{j=1}^{J}$$
$$\log \det \left( I + \sigma_{\text{obs}}^2 K(X_n^{+P_{F^{(j)}}^T(s)}, X_n^{+P_{F^{(j)}}^T(s)}) \right)$$
$$B(s; \infty|X_n) := \mathbb{E}_F$$
$$\left[ \log \det \left( I + \sigma_{\text{obs}}^2 K(X_n^{+P_F^T(s)}, X_n^{+P_F^T(s)}) \right) \right]$$

where $P_j^T(s)$ denotes the projected trajectory locations under the vector field sample $F^{(j)}$. Under these utilities, we would arrive at different optimal deployment locations, i.e. we would have

$$s_j^* := \operatorname{argmax}_s B(s; J|X_n),$$
$$s^* := \operatorname{argmax}_s B(s; \infty|X_n).$$

In addition, there is also the true optimal decision where we use the exact ground truth vector field $F_{\text{true}}$ to simulate the trajectories, i.e.

$$B(s; \text{true}|X_n) :=$$
$$\log \det \left( I + \sigma_{\text{obs}}^2 K(X_n^{+P_{\text{true}}^T(s)}, X_n^{+P_{\text{true}}^T(s)}) \right)$$

where $P_{\text{true}}^T$ are generated using $F_{\text{true}}$. This would give us the true optimal location $s_{\text{true}}^* := \operatorname{argmax}_s B(s; \text{true}|X_n)$.

To investigate the quality of the decision, as well as selecting the appropriate choice of $J$, we compute the utility gaps in the following two ways:

$$\text{Gap}_{\text{MC}}(J) := B(s^*; \infty|X_n) - B(s_J^*; \infty|X_n)$$
$$\approx B(s^*; 200|X_n) - B(s_J^*; 200|X_n),$$
$$\text{Gap}_{\text{Full}}(J) := B(s_{\text{true}}^*; \text{true}|X_n) - B(s_J^*; \text{true}|X_n).$$

The first gap is converging to zero as $J$ increases, whereas the second gap is not going to converge due to the difference between the probabilistic posterior model and the deterministic ground truth field.

### H.1.1. SYNTHETIC GROUND TRUTH

In this experiment, we will generate the ground truth field using a temporal Helmholtz GP model under the same specification as the synthetic ground truth experiment in the paper. The entire deployment spans $[0, 10]$. The decision times considered are $3, 5, 7$, and drifters are uniformly placed before the decision time every $0.5$ unit time.

In the plots, to put all values on the same scale, the percentage gaps, instead of raw gaps, are used, i.e.

$$\text{PercGap}_{\text{MC}}(J) := \text{Gap}_{\text{MC}}(J)/B(s^*; \infty | X_n) \times 100,$$
$$\text{PercGap}_{\text{Full}}(J) := \text{Gap}_{\text{Full}}(J)/B(s^*_{\text{true}}; \text{true} | X_n) \times 100.$$

Also, we consider two additional policies, Uniform and EIG, for comparison. The EIG policy looks at the location that maximises the expected information gain while considering only the initial deployment location (so no projection $P^T$). The uniform is selected uniformly from all the possible locations, which we compute using the average utility across the locations $s$.

In the result plot of Figure 8, the Monte Carlo gaps are shown on the first row, whereas the full gaps are shown on the second row. A horizontal line at 1 is added on the first row, with the corresponding $J$ values for the intersections added. We can see that after a rapid decay until around $J = 20$, the gap for BALLAST is reducing very slowly. The same happens for the second row with full gaps. We can also notice a superiority of BALLAST decisions over that of EIG and Uniform for almost all choices of sample number $J$.

### H.1.2. SUNTANS

We can also conduct a similar ablation study on the SUNTANS dataset, as outlined in Section 5.3. In Figure 9, we realise that setting $J = 20$ is reasonable, especially when considering the full stochasticity gap. The result in Section 5.3 is also reassuring on the quality of $J = 20$. We have also tried using $J = 100$, and the result is comparable to that of $J = 20$.

### H.2. Time Horizon Length

Another aspect of BALLAST is the time horizon for the forward projection of drifter trajectories. Algorithm 1 denotes the aggregated additional observations obtained after placing an observer at $s$ under the sampled field $F^{(j)}$ as $P_j^T(s)$, where $T$ - the terminal time of the full deployment - indicates the end time of forward projection for the sampled

field. Although it is natural to set the projection end time as $T$, using an earlier end time will reduce computational costs. In this ablation study, we investigate whether an earlier $T$ should be used.

The general setup is identical to that of Section H.1, and we only consider the Monte Carlo gap at decision time $t = 5$, without the loss of generality. In addition to the Uniform, EIG, and BALLAST decisions, we also consider BALLAST decisions with end time $T_{\text{end}} < T$, in particular, we have $T_{\text{end}} = 0.1, 0.5, 1, 2, 3$. The study is conducted using 100 different randomly generated ground truth vector fields to provide uncertainty quantification of the utility gaps.

As shown in Figure 10, the performance of BALLAST decisions increases uniformly as $T_{\text{end}}$ increases, suggesting that while computation allows, we should always set the end time as long as we can. Therefore, we will, as default, set $T_{\text{end}} = T$.

### H.3. Forward Simulation Discretisation Step Size

Discretisation of the forward trajectory We have now conducted an additional ablation study to investigate how sensitive the decision quality is to changing step sizes of forward projection. The paper considered stepsize = 0.05 as the default. We investigate the optimality gap of the decisions when the stepsize is 0.01 and 0.1 in the same setup as that of Section 5.1. Table 2 below reports the percentage utility gap (note: out of 100) of various step sizes using different numbers of posterior samples for decision time t=5. At the sample number J=20 (our recommended choice), all three stepsizes are well within $1\%$ optimal. Notice that although they vary and the smaller stepsizes yield a lower optimality gap, the improvement is mostly negligible, suggesting robustness of decision quality to forward projection approximations.

| Stepsize | $J = 20$ | $J = 60$ | $J = 100$ | $J = 140$ | $J = 180$ |
|---|---|---|---|---|---|
| 0.01 | 0.263 | 0.136 | 0.0906 | 0.0829 | 0.0851 |
| 0.05 | 0.187 | 0.274 | 0.189 | 0.177 | 0.155 |
| 0.10 | 0.243 | 0.338 | 0.296 | 0.281 | 0.248 |

*Table 2.* Percentage utility gap (%) for different forward projection stepsizes and posterior sample sizes. All tested stepsizes remain within $1\%$ optimality at $J = 20$, indicating robustness of decision quality to trajectory discretisation.

### H.4. Surrogate Model Choice

To isolate the gains of BALLAST from the surrogate model choice, we conducted an additional experiment under similar setup as Section 5.2 where an alternative surrogate model is used. In particular, we replaced the Helmholtz spatial kernel with an independent vector-output RBF kernel (called the velocity kernel in (Berlinghieri et al., 2023)) -- widely

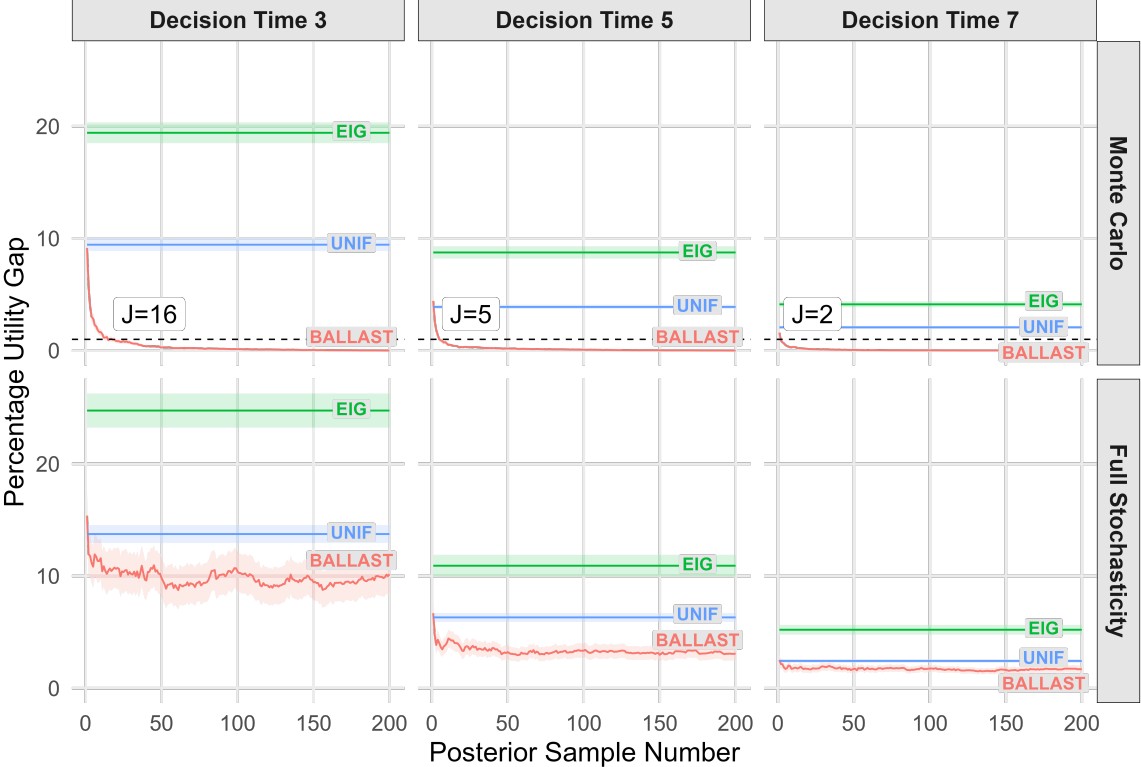

*Figure 8.* Combined plot of ablation under synthetic ground truth for decision times $t = 3, 5, 7$ with both Monte Carlo and full stochasticity percentage gaps. Three policies, EIG, Uniform, and BALLAST, are considered, and the two standard error bounds are shown.

considered suboptimal — and equipped EIG and BALLAST with such a misspecified surrogate. We compared SOBOL, BALLAST, EIG-mis and BALLAST-mis (with misspecified velocity kernel) and ranked their relative performance (1 = Best). As shown in the average rank Table 3, BALLAST-mis outperforms EIG-mis and even has similar performance to the SOBOL policy. While it is obvious that BALLST with a well-specified surrogate is the best performing policy here, our additional results provide further evidence that the gains we observed are due to the look-ahead construction of our proposed BALLAST algorithm.

| **Policy** | $n = 3$ | $n = 7$ | $n = 11$ | $n = 15$ | $n = 19$ |
|---|---|---|---|---|---|
| EIG-mis | 4.00 | 4.00 | 4.00 | 4.00 | 4.00 |
| BALLAST-mis | 1.70 | 1.90 | 2.00 | 2.25 | 2.10 |
| SOBOL | 2.25 | 2.10 | 2.20 | 2.10 | 2.10 |
| BALLAST | 2.05 | 2.00 | 1.80 | 1.65 | 1.80 |

*Table 3.* Average ranks (1 = best) of different policies under a misspecified surrogate model. BALLAST-mis consistently outperforms EIG-mis and remains competitive with SOBOL, highlighting the benefit of the BALLAST look-ahead strategy.

### H.5. Varying Flow Evolution Speed

To explore potential failure modes of BALLAST, we considered one such mode of the flow's evolution being too slow. For slowly evolving flows, we would expect the vector field to be almost stationary, in which case a space-filling design would be more desirable. To investigate the effect of varying flow speed, we considered changing the lengthscale of the temporal kernel from the present l = 2.5 (medium) to l = 5.0 (slow) as well as l = 0.5 (fast). Table 4 below shows the average rank of the performance of SOBOL, BALLAST, and UNIF for different flow speeds, with 1 = Best in the same setup as that of Section 5.2. We noticed that as the flow slows down, BALLAST starts to worsen and become on par with SOBOL, while it is much better than SOBOL for fast-flowing fields.

## I. Additional Experiment Details

Various experimental details of the investigations in Section 5 that are omitted or condensed in the main text due to space constraints are described here.

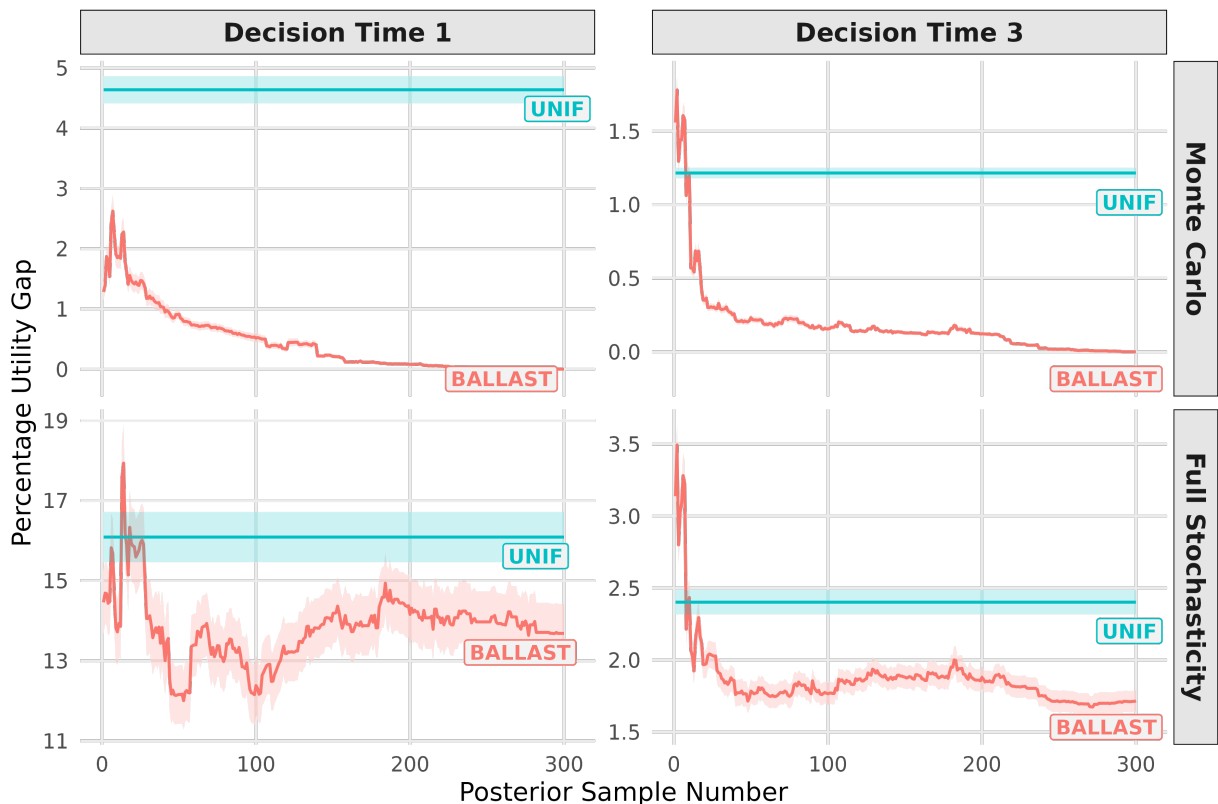

*Figure 9.* Combined plot of ablation under SUNTANS ground truth for decision times $t = 3, 5, 7$ with both Monte Carlo and full stochasticity percentage gaps. Three policies, EIG, Uniform, and BALLAST, are considered, and the two standard error bounds are shown.

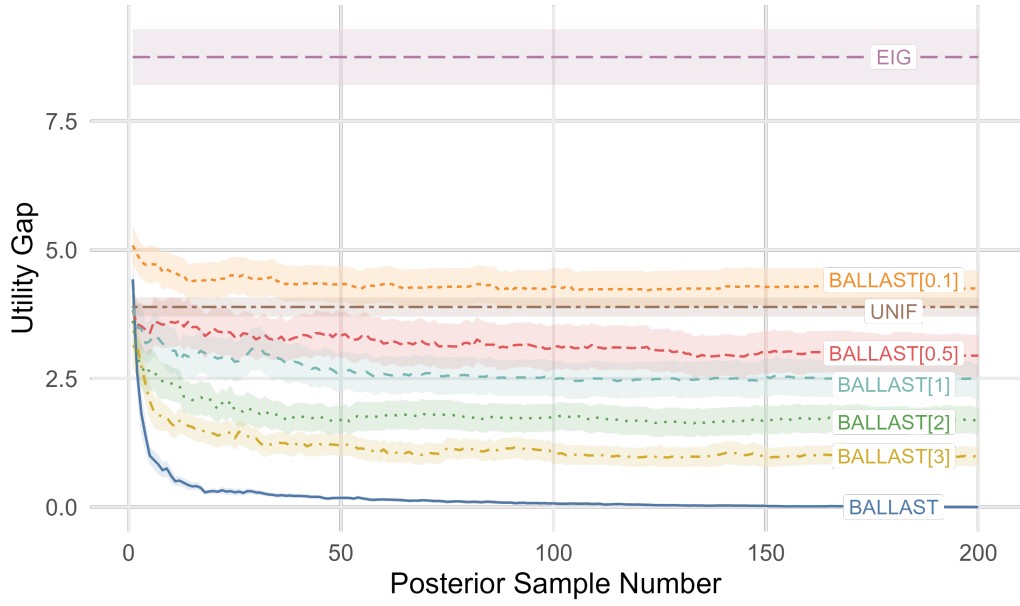

*Figure 10.* Utility gap with 2 standard error bounds of UNIF, EIG, and BALLAST decisions over sample number $J$ at decision times $t = 5$. BALLAST decisions with end times $T_{\text{end}} = 0.1, 0.5, 1, 2, 3$ are labelled as BALLAST$[T_{\text{end}}]$.

| Scenario | Policy | $n = 3$ | $n = 7$ | $n = 11$ | $n = 15$ | $n = 19$ |
|---|---|---|---|---|---|---|
| Fast | SOBOL | 2.10 | 2.45 | 2.30 | 2.15 | 2.25 |
| Fast | BALLAST | 1.70 | 1.05 | 1.15 | 1.05 | 1.05 |
| Medium | SOBOL | 1.55 | 1.55 | 1.60 | 1.65 | 1.60 |
| Medium | BALLAST | 1.45 | 1.45 | 1.40 | 1.35 | 1.40 |
| Slow | SOBOL | 1.55 | 1.60 | 1.50 | 1.45 | 1.45 |
| Slow | BALLAST | 1.45 | 1.40 | 1.50 | 1.55 | 1.55 |

*Table 4.* Average ranks (1 = best) of SOBOL and BALLAST under different flow speeds when comparing SOBOL, UNIF, and BALLAST. BALLAST performs substantially better for fast-evolving flows, while its advantage diminishes as the flow becomes slower.

### I.1. Observations from Lagrangian Observers

For a background time-dependent vector field $V(s, t)$ and a considered spatial region $R$, we simulate the trajectory of a Lagrangian observer initialised at time $t$ and location $s$ using Euler discretisation with a stepsize of $\delta_t = 0.01$, i.e. we have iterative updates

$$s_{n+1} = s_n + \delta_t V(s_n, t_n), \qquad t_{n+1} = t_n + \delta_t$$

for $n = 0, 1, \cdots$ with initial conditions $s_0 = s, t_0 = t$. We would also check if the observer has left the considered region each iteration, i.e. check $s \in R$, and terminate the update when it leaves, i.e. $s \notin R$. Furthermore, we only have access to the vector field $V(s, t)$ at the discretised spatial grid, so the velocity information within the same grid will be identical.

Given the underlying trajectory of an observer, we make observations at regular time intervals. In the experiments considered in Section 5, the observations are taken every $\delta_{\mathrm{obs}} = 0.05$ with additive i.i.d. Gaussian noise with standard deviation 0.1, so

$$y_k = V(s_k, t_k) + \varepsilon_k, \qquad \varepsilon_k \sim N(0, 0.1^2 I)$$

for observation $y_n$ at time $t_k$ and location $s_k$. Like above, the velocity information within the same grid of $V$ is set to be identical.

### I.2. Kernel Construction

BALLAST involves obtaining samples from the posterior GP at various sample times and locations. As described in Section 4.1, we first regress the observations using an extended GP $\boldsymbol{f} = [f, \partial_t f]^T$, then sample an initial condition for the posterior sample from it, which is then propagated via the SPDE approach.

In the temporal Helmholtz GP model we consider in this paper (see Section 2.1), the temporal component is set to be a separable Matérn $3/2$ kernel $k_t$. Implementing this model with the extended GP setup then requires double the output dimension for partial derivative values. One way of implementing the multi-output GP, such as the one considered here, is to augment

the input space with a binary indicator variable $z$, so the transformed scalar-output GP $g$ is constructed like $g(\cdot, z = 0) = f(\cdot)$ and $g(\cdot, z = 1) = \partial_k f(\cdot)$. See https://docs.jaxgaussianprocesses.com/_examples/oceanmodelling/ for an implemented example using GPJax of Pinder & Dodd (2022).

Next, we notice that many commonly used Python packages for GP implement the Matérn kernel with a clipped distance function. For example, GP-Jax implements the distance function of the kernel using `jnp.sqrt(jnp.maximum(jnp.sum((x - y) ** 2),1e-36))` to compute the distance between $x, y$. When taking the automatic hessian of the Matérn kernel implemented with the clipped distance at $x = y$, we would obtain 0 instead of the desired 3 (when $k_t$ is a Matérn $3/2$ with lengthscale 1 and variance 1). To bypass this issue, we should re-implement the Matérn kernel using `jnp.abs(x-y)` (for example) instead. Note that this only works for one-dimensional inputs $x, y$ - which is the case considered here.

### I.3. Considered Policies

The six policies considered in the experiments of Sections 5.2 and 5.3 are uniform (**UNIF**), Sobol sequence (**SOBOL**), distance-separation heuristic (**DIST-SEP**), **EIG**, BALLAST with optimised hyperparameters (**BALLAST-opt**) and BALLAST with true hyperparameters (**BALLAST-true**). Below, we will describe the details of the policies.

**UNIF** The UNIF policy draws uniformly a location from the spatial grid $R$ at each deployment.

**SOBOL** The SOBOL policy is implemented using Python's `scipy.stats.qmc.sobol` with scramble. We first generate points on the unit square $[0, 1)^2$, and then map it to our considered spatial grid $R$ where the points are converted into the indices of the spatial grid. Note that since we know the total number of deployments, the points are generated at once, which is not necessary as they can be produced sequentially. This policy is selected as a representative of space-filling designs such as Tukan et al. (2024).

**EIG**  The EIG policy implements the standard active learning of (2) where the GP model used is always identical to that of BALLAST-true.

**BALLAST-true**  The BALLAST-true policy implements Algorithm 1 where the GP hyperparameters are not optimised (i.e. Step 4 is skipped) but uses pre-determined, true values. The sample number $J$ is set to 20 following the ablation results in Sections 5.1 and H.1. The projection horizon is set to be the terminal time $T$ following the ablation result in Section H.2.

**BALLAST-opt**  The BALLAST-opt policy implements the full Algorithm 1 where the GP hyperparameters are optimised. The hyperparameters are estimated using the L-BFGS optimiser. Note that we impose manually-set bounds on the hyperparameter values during optimisation to mimic uniform priors with finite support. For the synthetic ground truth of Section 5.2, we set $[0.1, 1]$ bounds to all GP hyperparameters except for the temporal kernel, which we set $[0.1, 3]$. For the SUNTANS ground truth of Section 5.3, we set $[0.1, 5]$ bounds to stream kernel variance and potential kernel lengthscale, $[10, 20]$ bounds to time kernel variance and potential kernel variance, a $[0.1, 1]$ bound to stream kernel lengthscale, and a $[0.1, 3]$ bound to time kernel lengthscale. We should note that these bounds are set loosely to encourage the optimiser to stay within reasonable ranges. We have also observed that minor adjustments to the bounds do not change the results noticeably, as one would expect.

**DIST-SEP**  The deployment policy proposed by Chen et al. (2024b) works under the Lagrangian data assimilation inference framework, and considers two criteria: (1) "the drifters are deployed at locations where they can travel long distances within the given time window to collect more information about the flow field", and (2) "it is desirable to place the drifters at locations that are separate from each other".

These two criteria are computed using Lagrangian descriptors (Mancho et al., 2013; Chen et al., 2024a) in Chen et al. (2024b), which are obtained using a Monte Carlo average from posterior samples. As we are working under a different inference framework, we adapt their criteria and compute similar quantities for the two criteria using GP posteriors and BALLAST samples.

Here, we compute the criterion value for each point of the spatial grid. We compute the drifter length for each posterior sample (drawn exactly like BALLAST-true) by computing the total distance of sampled trajectories (the sum of the Euclidean distances between two consecutive observation locations). The separation is computed using the (negative) Euclidean distance between the potential deployment loca-

tion and the closest existing observation locations. Finally, we turn the values into index ranks and average the two ranks from the two criteria to make the final maximising decision.

The authors of Chen et al. (2024b) justified the two criteria in Section 4.2 by comparing them to the expected information gain policy. Therefore, it is not surprising to observe DIST-SEP performing worse than EIG in the experiments of Section 5.

### I.4. Computational Resources

The experiments of Section 5 are conducted on the SLURM computer cluster, where 100 CPUs are used for an embarrassingly parallel implementation of different starting seeds. Each job uses no more than 15GB of memory. The codes are implemented in Python 3.10, and mostly GPJax (Pinder & Dodd, 2022) version 0.11.0. The plots in the main texts are generated using either matplotlib in Python or ggplot2 in R.

