# OpenReview forum: "BALLAST: Bayesian Active Learning with Look-ahead Amendment for Sea-drifter Trajectories under Spatio-Temporal Vector Fields"
_ICML.cc/2026/Conference — ICML 2026 regular_

### Official Review · Reviewer_eNvM · 2026-02-21

**Soundness:** 3
**Presentation:** 4
**Significance:** 3
**Originality:** 3
**Overall Recommendation:** 5
**Confidence:** 3

**Summary:**

This paper proposes an active-learning deployment strategy for Lagrangian drifters with the consideration that sensors are advected by time-varying vector field. For each possible drop location, the proposed method will sample several likely flow fields from the current GP model, simulate, collect and score the location by how much those measurements are expected to reduce uncertainty.
The paper also introduces Vanilla-SPDE Exchange (VaSE) to improve the efficiency of drawing flow-field sample by using standard GP regression.

**Compliance With Llm Reviewing Policy:**

Affirmed.

**Key Questions For Authors:**

1. If you refine the spatial grid, do the chosen deployment locations change a lot, or are the improvements basically stable?

2. How robust is BALLAST when observations are irregular, delayed, or missing, instead of fixed δobs?

3. Is there a failure case where the look-ahead becomes misleading since posterior samples produce wrong trajectories?

**Limitations:**

1. Include stress-test or discussion on robustness. If the surrogate is misspecified and leads to incorrect flow, how bad will the look-ahead amendment become.

2. Add a short limitation subsection that list the key assumptions that may not hold in practice

**Strengths And Weaknesses:**

Strengths:
1. The problem formulation is on-point. They handle the key issue in the proposed area.

2. The look-ahead Amendment is clean. The formation and explanation are solid

3. VaSE is really useful. The paper identifies posterior field sampling as the main bottleneck and proposes the method to solve it.

4. The experimental evaluation is relatively comprehensive, and good ablation study to justify the design choices.

Weaknesses:
1. The proposed method relies on strong modeling assumptions. It is built around a specific GP surrogate for the flow and Gaussian noise.

2. The measurement model is a little bit idealized. In the experiments, it uses a fixed δobs=0.05 with i.i.d Gaussian noise. In real deployments, sampling could be more irregular, or even missing.

3. The trajectory simulation is quite discretization-heavy. That makes me worry the placement score could be partly driven by the discretization choices

---

> ### Author Rebuttal · Authors · 2026-03-30
>
> ## Discretisation of forward trajectory
>
> We have now conducted an additional ablation study to investigate how sensitive the decision quality is to changing stepsizes of forward projection. The paper considered stepsize = 0.05 as the default. We investigate the optimality gap of the decisions when the stepsize is 0.01 and 0.1 in the same setup as that of Section 5.1. The table below reports the percentage utility gap (note: out of 100) of various stepsizes using different numbers of posterior samples for decision time t=5. At the sample number J=20 (our recommended choice), all three stepsizes are well within 1% optimal. Notice that although they vary and the smaller stepsizes yield a lower optimality gap, the improvement is mostly negligible, suggesting robustness of decision quality to forward projection approximations.
>
> | Stepsize| J=20 | J=60 | J=100 | J=140 | J=180 |
> |---------|------|------|-------|-------|-------|
> | 0.01 | 0.263| 0.136| 0.0906| 0.0829| 0.0851|
> | 0.05 | 0.187| 0.274| 0.189 | 0.177 | 0.155 |
> | 0.10 | 0.243| 0.338| 0.296 | 0.281 | 0.248 |
>
> ## Deployment location and spatial grid
>
> The reviewer wondered how sensitive the decision on the grid size is, and if we could expect a stable improvement as the spatial grid gets refined. We have now explored this by considering how the decided location, and its corresponding utility, vary as the grid refines in the same setting as Section 5.2. In particular, we started with a 25*25 grid and considered the decision time t = 5.0 while using the same observations. We notice that as the grid gets refined, the location fluctuates slightly, although the corresponding utility remains stable, making them similarly viable choices of comparable qualities. This gives us evidence that the decision quality is insensitive to grid size choices.
>
> | Grid Number | Location | Utility |
> |-------------|----------------------|---------|
> | 20×20 | (-0.3158, 1.158) | 10.13 |
> | 25×25 | (0.000, 1.000) | 10.17 |
> | 30×30 | (-0.0690, 0.7586) | 10.16 |
> | 35×35 | (-0.2353, -0.2353) | 10.26 |
> | 40×40 | (-0.4615, -0.1538) | 10.22 |
>
> ## Measurement model
>
> We have now, for the rebuttal, conducted several additional studies to investigate the robustness of our method against various practical observational scenarios. We considered (a) observations with random missingness (equivalent also to irregular observations), and (b) observations with outliers. The random missingness is constructed by having an independently drawn Bernoulli random variable with a 0.1 success rate to determine if an additional observation is removed or not. The outliers are constructed by replacing the observation noise generation from an independent Gaussian random variable with a t-distributed (5 degrees of freedom) one of the same variance. The t-distribution has heavier tails than the Gaussian distribution, thus it encourages outliers in the observations.
>
> Under either of these two settings, we have observed that BALLAST withstands its generally superior performance against benchmarks. Below, we present the mean ranking (1 = Best)  when comparing SOBOL, BALLAST, and UNIF in the same setting as that of the experiment in Section 5.2 for varying numbers of placed drifters.
>
> | Scenario | Policy | n=3 | n=7 | n=11 | n=15 | n=19 |
> |----------|----------|------|------|------|------|------|
> | Outliers | SOBOL | 1.65 | 1.60 | 1.65 | 1.60 | 1.55 |
> | Outliers | BALLAST | 1.35 | 1.40 | 1.35 | 1.40 | 1.45 |
> | Missing | SOBOL | 1.65 | 1.40 | 1.50 | 1.45 | 1.50 |
> | Missing | BALLAST | 1.32 | 1.58 | 1.47 | 1.53 | 1.47 |
> | Standard | SOBOL | 1.55 | 1.55 | 1.60 | 1.65 | 1.60 |
> | Standard | BALLAST | 1.45 | 1.45 | 1.40 | 1.35 | 1.40 |
>
> ## Failure mode and Robustness to misspecified surrogate
>
> We refer the reviewer to our response to Reviewer 4fvy, especially the subsection with heading “Failure Mode” as well as the subsection with heading “Isolate gains from look-ahead and surrogate choice”.
>
> ## Discussion on assumptions
>
> We thank the reviewer for suggesting the inclusion of a limitation subsection. We have added that in the camera-ready version of the paper, discussing how real-world observations could break our modelling assumptions, as well as the potential misspecification of our surrogate model.

---

> > ### Author Rebuttal · Reviewer_eNvM · 2026-04-03
> >
> > Thank you to the authors for the detailed response. I have no further questions.

---

### Official Review · Reviewer_4fvy · 2026-03-13

**Soundness:** 2
**Presentation:** 3
**Significance:** 3
**Originality:** 3
**Overall Recommendation:** 4
**Confidence:** 3

**Summary:**

This paper studies active learning for placing Lagrangian observers to infer spatio-temporal vector fields. The main observation is that standard active learning utilities are suboptimal in this setting because they evaluate only the initial placement location and ignore the future trajectories of drifting observers. To address this, the paper proposes BALLAST, which estimates placement utility by simulating future trajectories under posterior vector-field samples, and VaSE, a GP inference procedure designed to make this posterior sampling more efficient. Experiments on both synthetic temporal Helmholtz data and a high-fidelity SUNTANS flow dataset show improved performance over several baselines.

**Compliance With Llm Reviewing Policy:**

Affirmed.

**Final Justification:**

All of my concerns have been adequately addressed, and I will maintain my score.

**Key Questions For Authors:**

Please refer to the Weaknesses section for the detailed context of my concerns. My main questions are about whether the reported gains come from the look-ahead mechanism itself or from the particular temporal Helmholtz surrogate, and about how robust the method is to modeling and numerical choices. Convincing responses on these points would likely improve my confidence in the paper.

**Limitations:**

No. The paper does not adequately discuss limitations or potential negative societal impact. It includes only a very brief impact statement that says there are many potential societal consequences but that none need to be specifically highlighted, and the main paper also lacks a dedicated discussion of practical failure modes or boundary conditions of the method.

**Strengths And Weaknesses:**

Strengths:
1. The motivation is explained clearly.
2. The derivations and technical presentation are fairly detailed.
3. The figures are generally clear and helpful.
4. The algorithmic presentation is concrete.

Weaknesses:
1. The paper does not isolate whether BALLAST's gains come from the look-ahead mechanism itself or from the temporal Helmholtz GP surrogate, and it does not test BALLAST with simpler or alternative surrogate families.
2. The paper lacks sensitivity analysis for the temporal kernel lengthscale/variance and the Helmholtz kernel's potential and stream lengthscales/variances.
3. VaSE's efficiency advantage is supported mainly by complexity analysis and order-of-magnitude estimates, without wall-clock or memory benchmarks.
4. BALLAST's forward projection relies on Euler rollout, grid-cell velocity assignment, and terminating observers once they leave the domain, but the paper does not analyze how these approximations affect decision quality.
5. The paper should further analyze regimes where BALLAST offers only marginal gains over simple space-filling policies, when it becomes close to the baseline, or when look-ahead projection may not be worth the extra computation.
6. The paper contains several typos and notation issues, such as "expect SOBOL", "polices" and "sexist".

---

> ### Author Rebuttal · Authors · 2026-03-30
>
> We thank the reviewer for the detailed reviews and for recognising that our paper is well-motivated, presents the derivations in detail, and the algorithm concretely.
>
> ## Isolate gains from look-ahead and surrogate choice
>
> The reviewer questioned whether the benefits we observed are from the surrogate model choice or the BALLAST look-ahead mechanism. Firstly, this is the same question we wish to answer by comparing EIG and BALLAST policies in the experiments – both share the same surrogate model, but BALLAST computes the EIG utility with look-aheads. And as we have repeatedly observed in the results in the paper, BALLAST trumps EIG noticeably.
>
> We have, now for this rebuttal, also conducted a new experiment to further explore this point. We replaced the Helmholtz spatial kernel with an independent vector-output RBF kernel (called the velocity kernel in [A])  – widely considered suboptimal – and equipped EIG and BALLAST with such a misspecified surrogate. We compared SOBOL, BALLAST, EIG-mis and BALLAST-mis (with misspecified velocity kernel) in the same setup as Section 5.2 and ranked their relative performance (1 = Best). As shown in the average rank table below, BALLAST-mis outperforms EIG-mis and even has similar performance to the SOBOL policy. While it is obvious that BALLST with a well-specified surrogate is the best performing policy here, our additional results provide further evidence that the gains we observed are due to the look-ahead construction of our proposed BALLAST algorithm.
>
> | Policy | n=3 | n=7 | n=11 | n=15 | n=19 |
> |--------------|------|------|------|------|------|
> | EIG-mis | 4.00 | 4.00 | 4.00 | 4.00 | 4.00 |
> | BALLAST-mis | 1.70 | 1.90 | 2.00 | 2.25 | 2.10 |
> | SOBOL | 2.25 | 2.10 | 2.20 | 2.10 | 2.10 |
> | BALLAST | 2.05 | 2.00 | 1.80 | 1.65 | 1.80 |
>
> ## Forward projection choices
>
> The reviewer questioned how the various forward projection “approximations affect decision quality”. We refer the reviewer to our reply to Reviewer eNvM, especially the subsection with the heading “Discretisation of forward trajectory”.
>
> ## Failure modes
>
> The reviewer suggested more discussions on the potential failure modes of our method. To this end, we considered one such failure mode of the flow’s evolution being too slow. For slowly evolving flows, we would expect the vector field to be almost stationary, in which case a space-filling design would be more desirable. To investigate the effect of varying flow speed, we considered changing the lengthscale of the temporal kernel from the present l = 2.5 (medium) to l = 5.0 (slow)  as well as l = 0.5 (fast). The table below shows the average rank of the performance of SOBOL, BALLAST, and UNIF for different flow speeds, with 1 = Best in the same setup as that of Section 5.2. We noticed that as the flow slows down, BALLAST starts to worsen and become on par with SOBOL, while it is much better than SOBOL for fast-flowing fields.
>
> | Scenario | Policy   | n=3  | n=7  | n=11 | n=15 | n=19 |
> |----------|----------|------|------|------|------|------|
> | Fast     | SOBOL    | 2.10 | 2.45 | 2.30 | 2.15 | 2.25 |
> | Fast     | BALLAST  | 1.70 | 1.05 | 1.15 | 1.05 | 1.05 |
> | Medium   | SOBOL    | 1.55 | 1.55 | 1.60 | 1.65 | 1.60 |
> | Medium   | BALLAST  | 1.45 | 1.45 | 1.40 | 1.35 | 1.40 |
> | Slow     | SOBOL    | 1.55 | 1.60 | 1.50 | 1.45 | 1.45 |
> | Slow     | BALLAST  | 1.45 | 1.40 | 1.50 | 1.55 | 1.55 |
>
> ## VaSE efficiency
>
> The reviewer requested further details on VaSE’s efficiency beyond computational costs. We measured the wall run times of VaSE on a consumer laptop equipped with an Apple Silicon CPU and 24 GB RAM. In the same setup as that of Section 5.2, drawing one posterior sample with VaSE takes under 4 seconds (3.77s at decision time t = 2.5, 3.89s at t = 5.0, 3.64s at t = 7.5), whereas SPDE takes around 4.5 minutes (245.77s at t = 2.5, 259.83s at t = 5.0, 281.75s at t = 7.5). This additional comparison provides further evidence that VaSE is noticeably more efficient than the state-of-the-art SPDE approach for the kinds of sampling tasks considered in the paper.
>
> ## Typos and societal impact
>
> We thank the reviewer for pointing them out. We have corrected the typos in the camera-ready version, as well as discussed our work’s potential societal impact more comprehensively (see below).
>
> This work presents an algorithm to optimise the placement of oceanographic drifters, with the primary aim of improving environmental monitoring of ocean dynamics. Enhanced observation of currents can contribute to more accurate climate models, a better understanding of marine ecosystems, and improved responses to environmental hazards such as oil spills or extreme weather events. Meanwhile, we acknowledge that similar methodologies could also be applied in industrial contexts, including oil and gas exploration and operations.
>
> [A] Berlinghieri, R., et al. (2023). Gaussian processes at the Helm (holtz): A more fluid model for ocean currents. ICML.

---

> > ### Author Rebuttal · Reviewer_4fvy · 2026-04-04
> >
> > Thank you for the rebuttal. My questions have been adequately addressed, and I have no further concerns at this point. I will maintain my current assessment.

---

### Official Review · Reviewer_MFrw · 2026-03-15

**Soundness:** 3
**Presentation:** 3
**Significance:** 3
**Originality:** 3
**Overall Recommendation:** 4
**Confidence:** 3

**Summary:**

The paper addresses the challenge of optimally placing sea-drifters (Lagrangian observers) to measure and map changing ocean currents over time. To solve this, the authors introduce a novel active learning framework, BALLAST, designed for Lagrangian observations using samples from surrogates. Experimental results also support their proposed hypothesis.

**Compliance With Llm Reviewing Policy:**

Affirmed.

**Key Questions For Authors:**

The active learning framework treats the deployment times of new drifters as fixed and predetermined, optimizing only their spatial locations rather than their deployment timing. This restriction simplifies the search space and reflects certain real-world operational constraints, but it also prevents the policy from adaptively choosing an optimal deployment schedule. Is it possible to extend the proposed approach to such a setting where an optimal schedule is important?

**Limitations:**

The method is evaluated using a high-fidelity numerical ocean model (SUNTANS), but it has not yet been validated in a real ocean deployment. In practice, live field experiments would introduce additional unmodeled effects—such as windage on surface buoys, sensor malfunctions, and drifter loss—that are not captured by these simulation-based tests. It remains unclear how the proposed approach would perform in those realistic environments.

Online computation costs are one of the main bottlenecks. The utility computation requires simulating many future trajectories using numerical ODE-solvers and aggregating expected information gain at every single decision step. Hence, running BALLAST online currently carries a heavy computational burden.

The optimization process involved might be sensitive or unstable without carefully specified human constraints, which might be a major roadblock to deploying this framework in varying environmental conditions.

**Strengths And Weaknesses:**

The problem is well formulated. The paper is also well written.

They proposed Vanilla SPDE Exchange (VaSE) for efficient posterior sampling, which is much more efficient (i.e., billions of times faster) than standard GP.

BALLAST is well evaluated against several baseline placement strategies. The experimental outcomes indicate that  BALLAST consistently outperformed the other methods. It achieved the same level of measurement accuracy while requiring significantly fewer drifters, translating to a deployment cost savings.

They model ocean currents using a specialized spatio-temporal Gaussian Process (GP) based on Helmholtz decomposition, which captures vector-field structures well.

The framework relies on the assumption that drifters transmit data instantly. Although modern buoys are fast, harsh ocean conditions can cause signal delays or dropouts, which would disrupt the algorithm's real-time planning.

The massive computational speedup achieved by the proposed Vanilla SPDE Exchange (VaSE) relies on specific structural assumptions about the surrogate model. This could be a bottleneck in scenarios where these assumptions fail.

---

> ### Author Rebuttal · Authors · 2026-03-30
>
> We thank the reviewer for the detailed reviews and for recognising that our paper addresses a well-formulated problem, is well-written, and the proposed BALLAST is well-evaluated against benchmarks.
> ## Relaxing the pre-determined deployment times
> We appreciate the reviewer's acknowledgement that the pre-determined deployment times assumed in our paper reflect real-world operational constraints. We sympathise with the desire to optimise deployment times and the deployment location. However, an unconstrained optimisation is unlikely to be informative and will add significant costs. Optimising the full set of deployments simultaneously is a very high-dimensional optimisation of the acquisition function. A similar problem arises in batch Bayesian optimisation, but most practical deployments use sequential greedy optimisation to alleviate the computational burden. We use the same trick here: each drifter deployment is optimised myopically, conditional on both the time of deployment and what has happened previously.
>
> We anticipate that for any drifter, it will be more informative to deploy as early as possible to maximise the number of observations. We could place a constraint in the optimisation to ensure sufficient time between deployments, but this becomes very problem-dependent. Furthermore, we anticipate deploying at the earliest feasible set of time points, which would directly lead to the setting we have presented – a pre-determined set of time points.
> ## Evaluations in more practical scenarios
> Surface currents are difficult to observe directly, particularly from satellite platforms, which typically retrieve only surface temperature, colour, and heights. So, comprehensive real-world spatio-temporal vector fields of surface current are scarce. The SUNTANS dataset used here is generated from a high-fidelity, three-dimensional hydrodynamic model, which is widely regarded in oceanography as producing physically realistic fine-scale circulation [A]. We also refer the reviewer to our response to Reviewer eNvM, especially the subsection with the heading “Measurement model” for a discussion on more realistic measurement models.
>
> ## Computational Time of BALLAST
>
> The reviewer worried that “running BALLAST online currently carries a heavy computational burden.” We conducted several runs and observed that the burden of BALLAST is light. For example, under the setup considered in Section 5.2, it consistently takes under 3 minutes (wall time, measured on a consumer laptop equipped with an Apple Silicon CPU and 24 GB RAM) to make a deployment decision with our recommended posterior sample number J = 20. In particular, at decision time t=3.0, it took 2 min 45 s; at decision time t=5.0, it took 1 min 55 s; at decision time t=7.0, it took 2 min 13 s.
>
> ## Surrogate Modelling and Optimisation
>
> The reviewer stated that “the massive computational speedup achieved by the proposed Vanilla SPDE Exchange (VaSE) relies on specific structural assumptions about the surrogate model. This could be a bottleneck in scenarios where these assumptions fail.” For VaSE to be applied, we only assume (1) a separable kernel in space and time, and (2) the temporal component admits the SDE reformulation. We believe that they are not overly restrictive: assumption 1 is standard in spatio-temporal modelling [C], and assumption 2 covers almost all the most common kernels (see Section 3.3 of [D]). Additionally, the Helmholtz spatial kernel considered in the paper is a common choice for ocean vector field modelling, as used by papers such as [E, F].
>
> Another concern is that “the optimization process involved might be sensitive or unstable without carefully specified human constraints”. The constraints on the kernel parameters we imposed in the optimisation process are commonly applied in Bayesian optimisation (e.g. Section 4 of [G]), not overly tweaked, and only provided a very rough, sensible parameter range for the optimiser to explore. Additionally, such prior information about the ocean flow is often available in practice.
>
> [A] Wang, B., et al. (2009) High-resolution simulations of a macrotidal estuary using SUNTANS. Ocean Modelling.
>
> [B] Masunaga, Eiji, et al. (2023) Numerical simulations of internal tide dynamics in a steep submarine canyon. Journal of Physical Oceanography.
>
> [C] ​​Hamelijnck, O., et al. (2021). Spatio-temporal variational Gaussian processes. NeurIPS.
>
> [D] Solin, A. (2016). Stochastic differential equation methods for spatio-temporal Gaussian process regression.
>
> [E] Ponte, A. L. S., et al. (2024). Inferring flow energy, space scales, and timescales: freely drifting vs. fixed-point observations. Nonlinear Processes in Geophysics, 31(4), 571-586.
>
> [F] Berlinghieri, R., et al. (2023). Gaussian processes at the Helm (holtz): A more fluid model for ocean currents. ICML.
>
> [G] Le Riche, R., & Picheny, V. (2021). Revisiting Bayesian optimization in the light of the COCO benchmark. Structural and Multidisciplinary Optimization.

---

> > ### Author Rebuttal · Reviewer_MFrw · 2026-04-05
> >
> > I thank the Authors for the response. Based on the response and the discussions with the other reviewers, I keep my score.

---

### Decision · Program_Chairs · 2026-04-30

**Decision:**

Accept (regular)

**Comment:**

This manuscript concerns an application of active learning for (quoting the authors) guiding the placement of Lagrangian observers to infer time-dependent vector fields.

The reception to this work during the initial review stage was positive, with reviewers praising the clarity of the presentation, the approach to modeling and problem formulation, and design of the empirical study. Although the reviewers did point out some relatively minor concerns, there was universal agreement among the reviewers that the paper was a solid contribution to the conference, and there was universal recommendation of acceptance.

I encourage the authors to take the feedback from the reviewers into account while revising the manuscript.